# Sub-diffractional cavity modes of terahertz hyperbolic phonon polaritons in tin oxide

Flávio H. Feres [1,2,9], Rafael A. Mayer[1,2,9], Lukas Wehmeier [3,4], Francisco C. B. Maia [1], E. R. Viana[5], Angelo Malachias[6], Hans A. Bechtel[7], J. Michael Klopf[8], Lukas M. Eng [3,4], Susanne C. Kehr[3], J. C. González[6], Raul O. Freitas [1✉] & Ingrid D. Barcelos [1✉]

Hyperbolic phonon polaritons have recently attracted considerable attention in nanophotonics mostly due to their intrinsic strong electromagnetic field confinement, ultraslow polariton group velocities, and long lifetimes. Here we introduce tin oxide ($SnO_2$) nanobelts as a photonic platform for the transport of surface and volume phonon polaritons in the mid- to far-infrared frequency range. This report brings a comprehensive description of the polaritonic properties of $SnO_2$ as a nanometer-sized dielectric and also as an engineered material in the form of a waveguide. By combining accelerator-based IR-THz sources (synchrotron and free-electron laser) with s-SNOM, we employed nanoscale far-infrared hyperspectral-imaging to uncover a Fabry–Perot cavity mechanism in $SnO_2$ nanobelts via direct detection of phonon-polariton standing waves. Our experimental findings are accurately supported by notable convergence between theory and numerical simulations. Thus, the $SnO_2$ is confirmed as a natural hyperbolic material with unique photonic properties essential for future applications involving subdiffractional light traffic and detection in the far-infrared range.

[1] Brazilian Synchrotron Light Laboratory (LNLS), Brazilian Center for Research in Energy and Materials (CNPEM), Campinas, SP, Brazil. [2] Physics Department, Gleb Wataghin Physics Institute, University of Campinas (Unicamp), Campinas, SP, Brazil. [3] Institute of Applied Physics, Technische Universität Dresden, Dresden, Germany. [4] ct.qmat, Dresden-Würzburg Cluster of Excellence-EXC 2147, Technische Universität Dresden, Dresden, Germany. [5] Department of Physics, Universidade Tecnológica Federal do Paraná (UTFPR), Curitiba, PR, Brazil. [6] Department of Physics, Universidade Federal de Minas Gerais (UFMG), Belo Horizonte, MG, Brazil. [7] Advanced Light Source (ALS), Lawrence Berkeley National Laboratory, Berkeley, CA, USA. [8] Institute of Radiation Physics, Helmholtz-Zentrum Dresden-Rossendorf, Dresden, Germany. [9] These authors contributed equally: Flávio H. Feres, Rafael A. Mayer. ✉email: raul.freitas@lnls.br; ingrid.barcelos@lnls.br

Phonon polaritons (PhPs) result from the coupling of electromagnetic fields and crystal lattice vibrations, creating bosonic quasi-particles analogous to photons that are confined at interfaces of the crystalline lattices having opposite signs of permittivity[1]. They exist from THz to mid-IR spectral frequencies, within Reststrahlen bands (RBs), situated between transversal $(\omega_{TO})$ and longitudinal $(\omega_{LO})$ optical phonon frequencies[2,3]. In nanostructured polar dielectric materials, PhPs enable confinement of light beyond the diffraction limit[3,4] allowing super-resolution imaging[5], thermal emission[6], data storage[7] and offer several advantages, mainly related to the usual higher quality factors and significant lower optical losses[8] of PhPs compared to plasmon polaritons[9]. Accordingly, PhPs are regarded as an essential element in modern applications such as molecular sensing[10], subdiffractional waveguiding[11], nano-resonators[12] and phonon-enhanced microscopy[13], primarily in the IR spectral range. Particularly in anisotropic media, in which the permittivity tensor $\overleftrightarrow{\varepsilon}$ possesses both positive and negative principal components, PhPs propagate inside the material (volume-confined) and exhibit hyperbolic dispersion[14]. Consequently, hyperbolic 2D materials emerge as robust platforms to study nanoscale light–matter interactions as well as fundamental building blocks for future nanophotonics[15,16].

In the mid-infrared (IR), strongly confined hyperbolic phonon polaritons (HPhPs) in hBN[12,17] and α-MoO$_3$[18–20] have gained much attention as they exhibit natural hyperbolicity and, hence, enhanced waveguiding properties. Consequently, the search for quantum materials that can support HPhPs in alternative energy ranges is of considerable interest. In the photonics scope, SnO$_2$ shows negative permittivity in specific spectral ranges, from mid- to far-IR, where different types of polaritons coexist. Accordingly,

SnO$_2$ serves as a unique platform to study the optical transport of multimode PhPs. The compelling polaritonic properties of SnO$_2$ stem from its crystalline structure (Fig. 1a) leading to an anisotropic phononic resonant $\overleftrightarrow{\varepsilon}$, which is in-plane isotropic $(\varepsilon_{xx} = \varepsilon_{yy})$ and out-of-plane anisotropic $(\varepsilon_{zz} \neq \varepsilon_{xx}, \varepsilon_{yy})$. The inversion of signs of the real parts of the permittivity components in different RBs inside the mid- and far-IR ranges configures a hyperbolic dispersion for SnO$_2$. Moreover, polar nanometer-sized crystals of SnO$_2$ ribbons or nanobelts (SnO$_2$-NBs)[21–23] have been reported as key elements in gas sensors[24], solar cells[25], lithiation electrodes[26–28], photonic devices[29], flexible and transparent electrodes[30], water treatment catalysts[31], electrochemically active layers in hydrogen peroxide production[32], and photocatalysis[33]. This is mainly due to their unique optical[21,24] and electronic[34,35] properties and large surface to volume ratio.

Here we present SnO$_2$-NBs as a lithography-free nanophotonic platform suitable for cavity confinement of far-IR HPhPs. Assisted by scattering scanning near-field optical microscopy (s-SNOM)[36–38] coupled to accelerator-based sources (Fig. 1e, f), we employed broadband IR-THz Synchrotron Infrared Nanospectroscopy (SINS)[39–41] and IR-THz free-electron laser s-SNOM narrowband imaging (FEL s-SNOM)[42–44] to experimentally access HPhP cavity modes in SnO$_2$-NBs by direct nano-imaging of transverse HPhP standing waves. Our experimental observations are comprehensively described and supported by numerical simulations as well as analytical modeling for the NB as a Fabry–Perot (FP) cavity. Overall, our findings expand the possibilities of SnO$_2$-NBs from an established 1D-semiconductor to a unique multimode hyperbolic material naturally optimized for the realization of subdiffractional resonators and, potentially, waveguiding in the far-IR range.

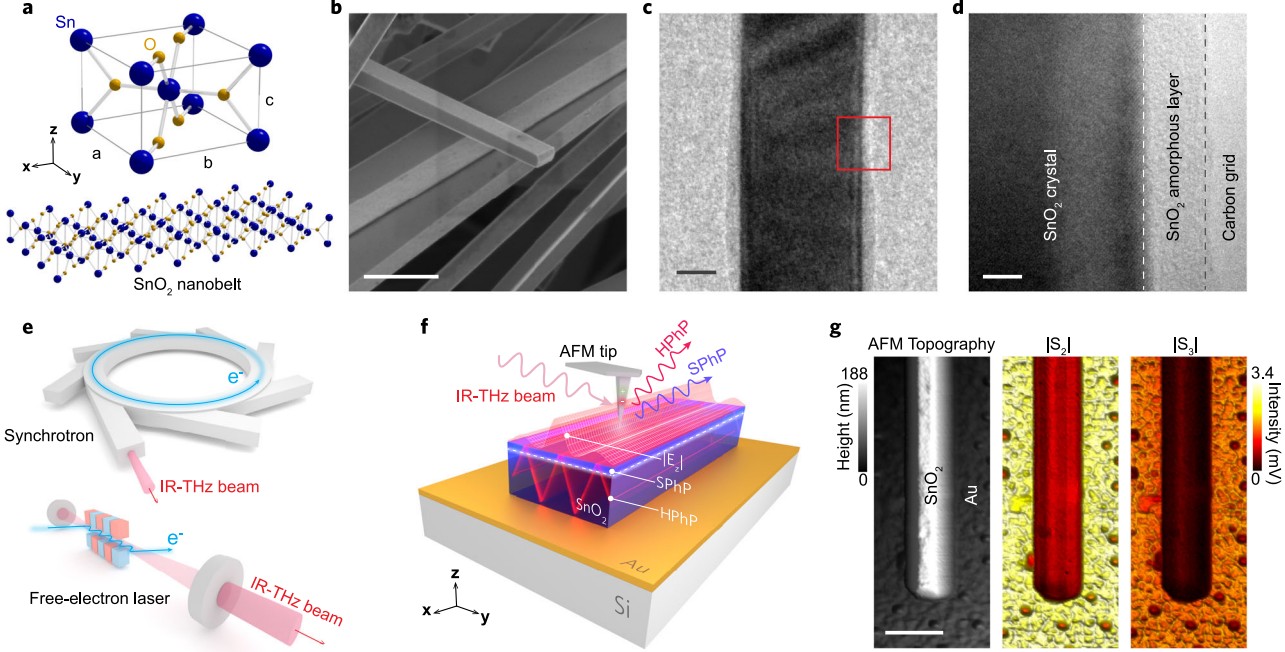

**Fig. 1 Overview of SnO$_2$-NBs morphology and spectral nano-imaging experiments. a** Schematic of SnO$_2$ unit cell for tetragonal Rutile, and crystalline structure of the nanobelt. Blue and yellow spheres represent tin (Sn) and oxygen (O) atoms, respectively. **b** SEM image of SnO$_2$-NBs. **c** TEM image of an isolated nanobelt covered by a thin amorphous layer in a carbon grid. **d** High-magnification TEM image (red square in **c**) highlighting the crystalline structure covered by a thin amorphous layer. **e** Accelerator-based IR-THz sources employed in the spectral nano-imaging experiments. **f** Experimental schematic showing the IR-THz beam illuminating a metallic AFM tip (nano-antenna) for the s-SNOM experiment. The highly confined and vertically polarized electric fields ($E_z$) at the tip apex launch surface (SPhPs) and volume (HPhPs) polaritons waves in the SnO$_2$-NB. **g** Morphology (AFM topography) and broadband reflectivity ($|S_2|$ and $|S_3|$) nanoscale images of an isolated SnO$_2$-NB/Au simultaneously measured by SINS. Scale bars in **b**, **c**, **d**, and **g** represent 1 μm, 50 nm, 5 nm, and 500 nm, respectively.

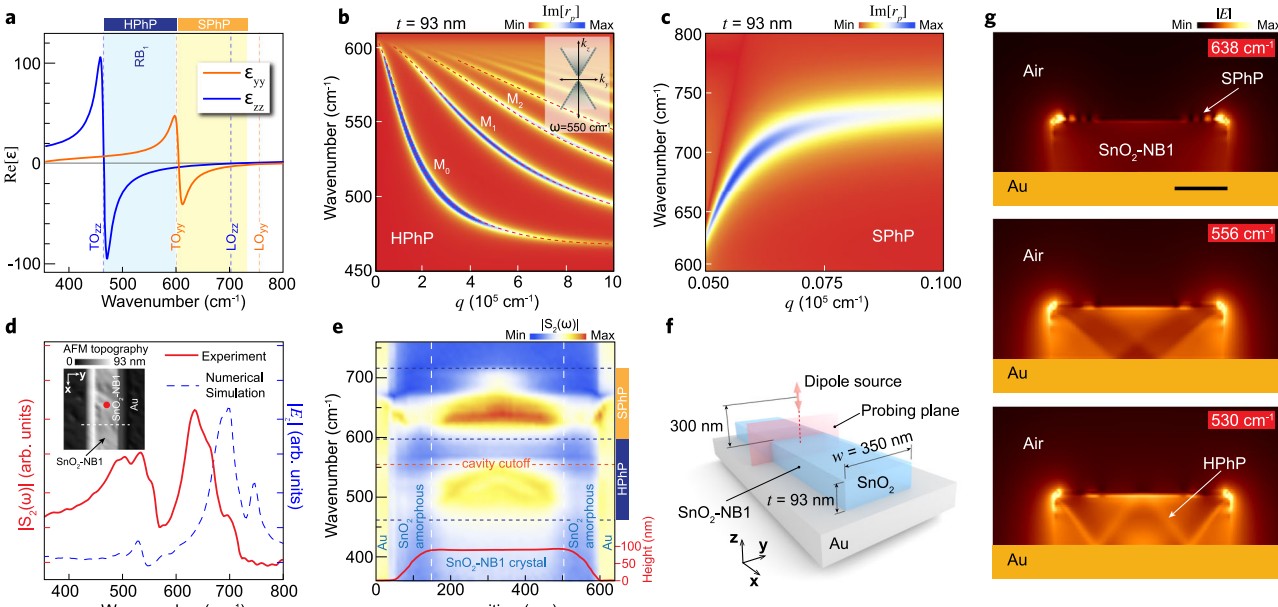

**Fig. 2 Polaritonic activity in SnO₂-NBs. a** Real part components of the electrical permittivity ($\varepsilon_{yy}$ and $\varepsilon_{zz}$) of SnO₂ showing the hyperbolic (HPhP) spectral region (light blue-shaded Reststrahlen band type I, RB₁) and surface phonon polaritons (SPhP) window (light yellow-shaded). **b**, **c** Dispersion relation for volume (HPhPs) and surface (SPhPs) phonon polaritons in SnO₂, respectively. False-color plot represents the calculated imaginary part of the complex reflectivity Im[$r_p$] for an air/SnO₂/Au multilayered structure. Dashed lines in **b** represent the Im[$r_p$] extracted from Eq. 2. $M_0$ and $M_{1,2,...}$ represent the fundamental and high order phonon mode dispersions, respectively. Inset **b** depicts the dispersion in isofrequency at 550 cm⁻¹. **d** SINS amplitude $S_2(\omega)$ point spectrum (red solid line) of the SnO₂-NB1. Inset shows 1×1 μm² AFM topography of the SnO₂-NB1 ($t = 93$ nm, $w = 350$ nm), and the red dot indicates the SINS point spectrum location. FDTD-simulated spectrum (blue dashed line) obtained from |$E_z$| integration along the red-dashed vertical line in **f**. **e** SINS spectral linescan along the white-dashed line in the inset **d**. The red solid line profile at the bottom represents the AFM height of the SnO₂-NB1. Vertical white-dashed lines delimit amorphous and crystalline SnO₂ phases. Horizontal blue dark-dashed lines denote the HPhPs and SPhPs spectral ranges. The horizontal orange dashed line indicates the cutoff frequency for this NB. **f** Parameters-space for the FDTD numerical simulations of the SnO₂-NB1 sample. **g** Simulated electric field |E| distribution inside the NB revealed by cross-sections at the frequencies 638 (outside RB₁), 556, and 530 cm⁻¹, respectively. The |E| intensity cross-sections highlight the presence of SPhPs (outside RB₁) and volume standing waves (HPhPs inside RB₁).

## Results and discussion

**Morphology, atomic structure, and broadband nano-reflectivity of SnO₂-NBs.** Tetragonal Rutile SnO₂ (P4₂/mnm, space group 136) with lattice constants $a = b = 0.473$ nm and $c = 0.318$ nm in the form of NBs (Fig. 1a) were morphologically characterized by scanning and transmission electron microscopy (SEM and TEM). Figure 1b shows a SEM image of a set of NBs with a clear view of the smooth surface quality and average rectangular shape of the NBs. Typical transverse dimensions are 50–500 nm wide ($w$) and a few hundreds of nm thick ($t$). Lengths can reach up to 50 μm. Figure 1c presents a TEM image of a 120 nm wide isolated SnO₂-NB. Figure 1d displays a high-magnification TEM image of the side edge of the NB (red square area in Fig. 1c) with a clear contrast between crystalline and amorphous phases of the nanostructure[34]. Complementary, SINS broadband imaging (Fig. 1g) unveils morphology (AFM topography) and broadband local reflectivity of a SnO₂-NB transferred to Au substrate, a standard configuration for all samples analyzed in this work. The AFM topography corroborates the SEM and TEM morphological analysis, while the s-SNOM amplitude maps highlight broadband reflectivity indicating a preliminary and qualitative view of optical confinement in the NB. |$S_2$| and |$S_3$| represent the 2nd and 3rd harmonics of the s-SNOM tip demodulation, respectively, and confirm the high signal-to-noise ratio and background-free quality of the analysis. This work analyzed three NBs samples with different dimensions ($t$, $w$): SnO₂-NB1 (93 nm, 350 nm), SnO₂-NB2 (130 nm, 200 nm), and SnO₂-NB3 (120 nm, 700 nm).

**Multimode PhPs in SnO₂-NBs: theory and experimental assessments.** The SnO₂ Rutile-type structure is known to exhibit optical phonons in the mid- and far-IR frequency ranges[45,46]. Thus, the knowledge of these specific phonon frequencies is key to define the photonic suitability of materials. The SnO₂ polaritonic properties are defined in terms of the phononic resonant $\overset{\leftrightarrow}{\varepsilon}$ components expressed in the Lorentz model

$$\varepsilon_\beta = \varepsilon_{\beta,\infty} \left( 1 + \sum_j \frac{\left(\omega_{LO,j}^\beta\right)^2 - \left(\omega_{TO,j}^\beta\right)^2}{\left(\omega_{TO,j}^\beta\right)^2 - \omega^2 - i\omega\Gamma_j^\beta} \right) \quad (1)$$

where $\beta$ denotes the $\overset{\leftrightarrow}{\varepsilon}$ component, with $\beta = xx = yy$ or $zz$, having a number $j$ of active optical phonons. $\varepsilon_{\beta,\infty}$ is the high-frequency term permittivity, $\omega$ is the excitation frequency and $\Gamma_j^\beta$, the dielectric damping. Using specific values for each of those parameters (Supplementary Table 1) we obtain the optical response of the SnO₂ as presented in Fig. 2a. Vertical dashed lines indicate the frequencies for SnO₂ transverse and longitudinal optical phonon modes in-plane ($yy$) and out-of-plane ($zz$), whose values are 605 cm⁻¹ (TO$_{yy}$), 757 cm⁻¹ (LO$_{yy}$), 465 cm⁻¹ (TO$_{zz}$), and 704 cm⁻¹ (LO$_{zz}$).

These phonon modes give rise to the RB, where the real parts of the in-plane and out-of-plane permittivities exhibit opposite signs (Re[$\varepsilon_{yy}$] · Re[$\varepsilon_{zz}$] < 0), indicating that the SnO₂ is in the class of material with a hyperbolic isofrequency surface, described by

$\frac{k_z^2}{\varepsilon_{yy}} + \frac{k_y^2}{\varepsilon_{zz}} = k_0^2$. Specifically, the hyperbolic window (shaded in light blue in Fig. 2a), defined here as RB type I (RB$_1$) with Re[$\varepsilon_{yy}$] > 0, and Re[$\varepsilon_{zz}$] < 0, is delimited in the frequency range 465–605 cm$^{-1}$. In this case, these modes only exist inside the crystal volume and propagate with a well-known angle $\theta_v(\omega) = \tan^{-1}\left[ i\sqrt{\frac{\varepsilon_{yy}}{\varepsilon_{zz}}} \right]$ with respect to the $z$ axis. Although this is a general description for the SnO$_2$, our approach is limited to thin films as we explore only ultra-confined phenomena, $k_y \gg k_0$, with $k_y = q + i\kappa$ being the complex in-plane momentum of the HPhP waves.

Figure 2b, c displays the calculated frequency–momentum ($\omega - q$) PhPs dispersion relation for a 2D SnO$_2$ flake ($t = 93$ nm). The false-color maps feature the imaginary part of the Fresnel reflectivity coefficient (Im[$r_p$]) of incident p-polarized light by a multilayered system comprising the Au substrate, SnO$_2$ hyperbolic medium, and air. $r_p$ is achieved through the following equation:

$$r_p = \frac{r_a - r_s e^{i2k_{ez}t}}{1 + r_a r_s e^{i2k_{ez}t}} \tag{2}$$

where $r_a = \frac{\varepsilon_{yy}k_{air} - \varepsilon_{air}k_{ez}}{\varepsilon_{yy}k_{air} + \varepsilon_{air}k_{ez}}$ and $r_s = \frac{\varepsilon_{Au}k_{ez} - \varepsilon_{yy}k_{Au}}{\varepsilon_{Au}k_{ez} + \varepsilon_{yy}k_{Au}}$ represent the reflectivity coefficients at the interfaces air and Au substrate, respectively. $t$ represents the thickness of the SnO$_2$ slab. Those equations also take into account the $z$ axis momentum $k_i = \sqrt{\varepsilon_i k_0^2 - k_y^2}$, for each medium $i$ (where $i = $ Au or air) and the extraordinary momentum inside the anisotropic medium, $k_{ez} = \sqrt{\varepsilon_{yy}k_0^2 - \frac{\varepsilon_{yy}}{\varepsilon_{zz}}k_y^2}$. $\varepsilon_{Au}$ and $\varepsilon_{air}$ are the Au and air permittivities, respectively.

For the hyperbolic modes, considering highly confined subdiffractional waves ($k_y \gg k_0$), we can rewrite the $z$ axis and extraordinary momentum as $k_i = ik_y$ and $k_{ez} = i\sqrt{\frac{\varepsilon_{yy}}{\varepsilon_{zz}}}k_y$. Thus, an analytical expression can be obtained directly from the relation, $1 + r_a r_s e^{i2k_{ez}t} = 0$. Consequently, this simplified dispersion relation, described by Eq. 3, allows for a more trivial mathematical handling[17], where $i\pi l$ represents the multiple branch solutions and the signs ($\mp$) follows the dispersion slope.

$$k_y(\omega) = q(\omega) + i\kappa(\omega) = \frac{1}{2t}\sqrt{\frac{\varepsilon_{yy}}{\varepsilon_{zz}}}\left[ \ln\left( \frac{\varepsilon_{yy} - \varepsilon_{air}\sqrt{\frac{\varepsilon_{yy}}{\varepsilon_{zz}}}}{\varepsilon_{yy} + \varepsilon_{air}\sqrt{\frac{\varepsilon_{yy}}{\varepsilon_{zz}}}} \right) \right.$$
$$\left. + \ln\left( \frac{\varepsilon_{Au}\sqrt{\frac{\varepsilon_{yy}}{\varepsilon_{zz}}} - \varepsilon_{yy}}{\varepsilon_{Au}\sqrt{\frac{\varepsilon_{yy}}{\varepsilon_{zz}}} + \varepsilon_{yy}} \right) \mp i\pi l \right], l = 1, 3, 5\ldots \tag{3}$$

In stark contrast, a major difference between HPhPs and surface phonon polaritons (SPhPs) is revealed by their dispersions (Fig. 2b, c) and consequently their $\theta_v$. Surface phonons cannot exist inside the volume, hence, $\theta_v$ assumes pure imaginary values by the given condition: Re[$\varepsilon_{zz}$] < 0 and Re[$\varepsilon_{yy}$] < 0. This indicates that these modes are confined to surfaces and interfaces with a single branch and do not form standing waves. Yet, in the hyperbolic spectral region, multiple distinct branches of HPhPs can exist. The fundamental branch ($l = 1$) is defined as $M_0$ and is the mode accessed experimentally in this work. The multiple distinct branches $M_m$ are defined for $m = 0, 1, 2, 3, \ldots$ and, hence, from branches $l = 2m + 1$ (Eq. 3). Fundamental ($M_0$) and high order ($M_1, M_2, \ldots$) modes dispersions are displayed in Fig. 2b inside the RB$_1$, indicating that multimode HPhPs can propagate inside SnO$_2$-NBs. Moreover, the branches slope ($\partial\omega/\partial q$) inside the RB$_1$ have similar behavior to the RB$_1$ of hBN slabs[47]. Finally, the inset of Fig. 2b shows an isofrequency diagram for a SnO$_2$ RB$_1$ calculated at 550 cm$^{-1}$.

We employed SINS to experimentally access the full spatial-spectral response and imaging of PhPs in SnO$_2$-NBs. Free-space broadband mid- to far-IR synchrotron radiation is strongly confined at the apex of a metallic AFM tip (see "Methods" section) allowing the launching and detection of PhPs waves in SnO$_2$-NBs (Fig. 1f), enabled by the momentum match between the s-SNOM source/probe and these quasi-particles. The full spectral response across the RB$_1$ of the sample SnO$_2$-NB1 is presented in Fig. 2d. The SINS point spectrum, acquired at the center of the NB (red dot in Fig. 2d AFM inset), unveils strong IR activity in the frequency range from 420 to 700 cm$^{-1}$, which are assigned to Sn–O antisymmetric vibrations. Within that spectral range, the peak at 686 cm$^{-1}$ is assigned to Sn–O–Sn vibrations, and the bands in the lower frequency range (430–620 cm$^{-1}$) are attributed to Sn–O stretching vibrations[48,49].

To explore the polaritonic response of the SnO$_2$-NB1, we acquired a spectral linescan with 25 nm spatial resolution (Fig. 2e) transverse to the NB (white-dashed line in Fig. 2d inset). Each point over the line carries a normalized amplitude spectrum ($|S_2|$ = $|S_{SnO2}|/|S_{Au}|$), as organized in the distance-frequency map displayed in Fig. 2e. An AFM profile (red line at the bottom) overlaps the $|S_2|$ spatio-spectral map allowing for a direct correlation between morphology and optical response of the NB. The AFM profile confirms the dimensions 93 nm × 350 nm ($t$ × $w$) for the SnO$_2$-NB1 sample. The linescan experiment unveils a complex spatial-frequency $|S_2|$ intensity pattern within the crystalline phase of the SnO$_2$-NB, delimited by vertical white-dashed lines. The spectral response observed is consistent with the calculated permittivity and dispersion relations in Fig. 2a–c. In the range above 600 cm$^{-1}$, a strong and uniform near-field response is attributed to SPhPs, as predicted from the single branch dispersion relation in Fig. 2c. Within the RB$_1$ range, the near-field intensity fringes produce an interference pattern of frequency-dependent standing waves, suggesting transverse volume confinement of HPhP waves inside the NB. The horizontal orange dashed line in Fig. 2e indicates the cutoff frequency for the SnO$_2$-NB1, further discussed in this report.

To support the interpretation of the SINS spatio-spectral analysis, we employed finite-difference time-domain (FDTD) numerical simulations to estimate the mid- to far-IR near-field response of the SnO$_2$-NB. Figure 2f presents the parameter space for the numerical simulation, where the metallic tip was modeled as an ideal dipole source positioned 300 nm above the NB surface (see "Methods" section). A SnO$_2$ nanobelt with dimensions $t$ = 93 nm, $w$ = 350 nm, and infinite length was defined as the polar crystal, analogous to the real morphology of the SnO$_2$-NB1 sample. The substrate was modeled as Au in accordance with the experimental conditions. Figure 2d shows a simulated $|E_z|$ spectrum (blue dashed amplitude profile) reconstructed from the integration of the out-of-plane electric field underneath the model dipole (red-dashed line in Fig. 2f) at the center of the NB. By comparing experiment and numerical simulation in Fig. 2d, a qualitative match is noticed for the central frequency of the main spectral features. In the line shape comparison, there is a fair correspondence between experiment and modeling for the SPhPs (peaks above 600 cm$^{-1}$), while the HPhP range appears to be less trivial to model since volume waves are highly sensitive to the morphology of the NB. The divergence between simulated and measured spectrum in Fig. 2d can be attributed to realistic experimental aspects not taken into account in the simulation (see "Methods" section). For a better understanding of the volume PhPs in SnO$_2$-NBs, Fig. 2g displays simulated electric field intensity maps from cross-sections of the SnO$_2$-NB1 (probing plane in Fig. 2f) when excited by a broadband dipole source (see "Methods" section). At 638 cm$^{-1}$ (inside the SPhPs window) there is no evidence of the polaritonic activity below the

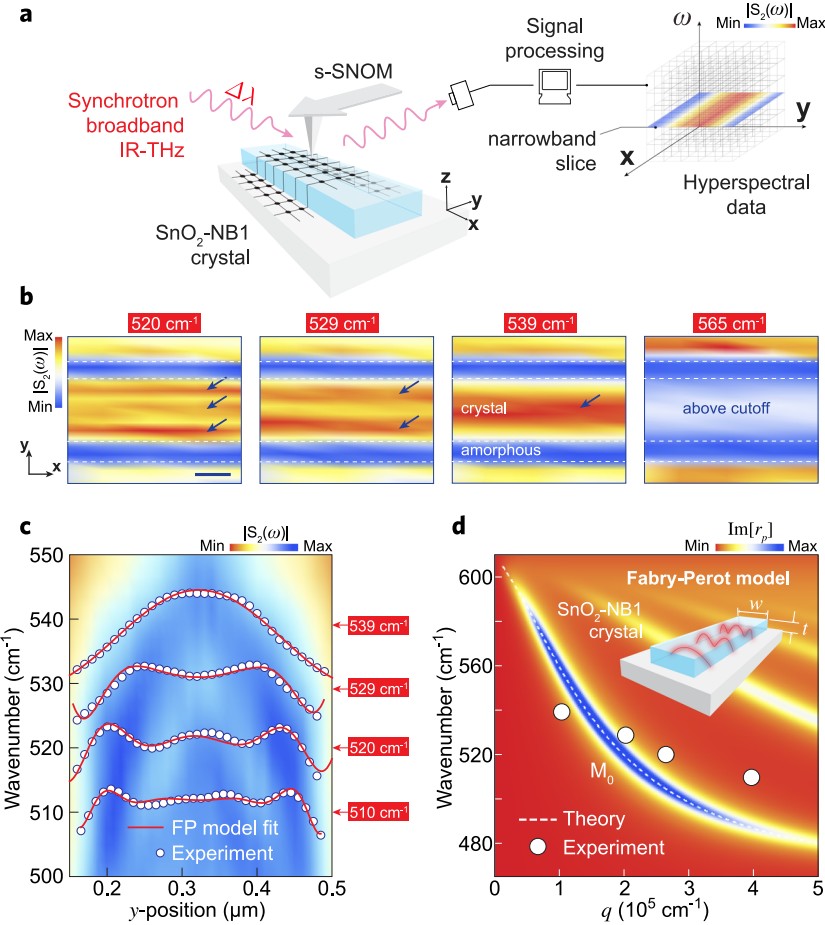

**Fig. 3 Experimental visualization of cavity modes in SnO₂-NBs. a** Schematic of the SINS experiment for broadband IR-THz (Δλ) hyperspectral imaging. **b** SINS amplitude S₂(ω) narrowband maps (10 cm⁻¹ spectral width) reconstructed from a full hyperspectral image of SnO₂-NB1/Au for the frequencies 520, 529, 539, and 565 cm⁻¹. Horizontal white-dashed lines delimit crystalline and amorphous phases of the NB. Blue arrows guide the eyes highlighting the interference fringes formed by HPhP waves inside the SnO₂-NB1 crystal. Scale bar represents 200 nm. **c** Spectral linescan (false-color map) superposed by SINS amplitude profiles (white circles) extracted from Fig. 2e. Red solid lines represent the FP model fitting (Eq. 5) of the experimental profiles at 510, 520, 529, and 539 cm⁻¹. Amplitude profiles are in arbitrary units and were vertically offset for clarity. **d** Dispersion relation for HPhP volume fundamental mode (M₀) calculated analytically from Eq. 2 through the imaginary part of the complex reflectivity Im[rₚ] (white-dashed line). White circles represent frequency–momentum (ω–q) values model-extracted from the fittings in **c**. Inset illustrates the cavity modes (FP model) for a NB with thickness t and width w.

surface, while in the HPhP window we can clearly notice volume modes at 556 and 530 cm⁻¹. The polaritonic response observed in the SINS spectral linescan (Fig. 2e) and supported by numerical simulations (Fig. 2g) provides a clear indication of the existence of HPhPs standing waves inside the polar crystal, therefore, configuring the SnO₂-NB as a transverse FP cavity for PhPs.

We carried out SINS hyperspectral (HS) imaging (see Supplementary Note 2) of the SnO₂-NB1 for a real-space visualization of the HPhP cavity modes. Figure 3b presents a series of frequency-selected nano-images inside the SnO₂ RB₁. These narrowband images were reconstructed from a full HS map in which each pixel of a 2D scanned image contains a full SINS spectrum, as described by the experimental diagram in Fig. 3a. These images covered a full cross-section of the NB enabling the visualization of interference fringes inside the crystal that are strongly dependent on the excitation frequency. The images at 539, 529, and 520 cm⁻¹ unveil a systematic increase in the number of intensity maxima inside the NB (blue arrows in Fig. 3b), while the image at 565 cm⁻¹ displays a flat and relatively weak near-field response, since this last excitation frequency lies well above the cutoff for this cavity (555 cm⁻¹, from Fig. 2e). In

contrast to dielectric waveguides where the cutoff frequency is at the lower limit of the waveguide mode dispersion, for the SnO₂-NBs the cutoff frequency is at the upper limit since the magnitude of the transversal wavevector increases with decreasing frequency, as a consequence of the band type I dispersion of the SnO₂ HPhP modes. Therefore, the HS analysis provides direct experimental evidence for the presence of HPhP cavity modes inside the crystalline phase of the SnO₂-NBs.

Figure 3c shows a spectral linescan (false-color map) of the SnO₂-NB1 and respective amplitude profiles extracted at frequencies 510, 520, 529, and 539 cm⁻¹. In order to understand the confining mechanism that regulates the wave patterns observed in Fig. 3b, c, we considered a model where the tip-launched HPhP waves (M₀ mode), traveling across the y axis of NB (Fig. 3a), are reflected by its side edges forming a FP cavity (inset Fig. 3d). The mode M₀ accumulates a round-trip phase (left part of the Eq. 4) that is equal to multiple integers of 2π (right part of the Eq. 4), thus, satisfying the FP maxima interference condition according to the following equation[50,51]:

$$2q_{M0}w + 2\varphi = (n+1)2\pi \qquad (4)$$

where $w$ is the width of the NB, $\varphi$ is the phase acquired in the

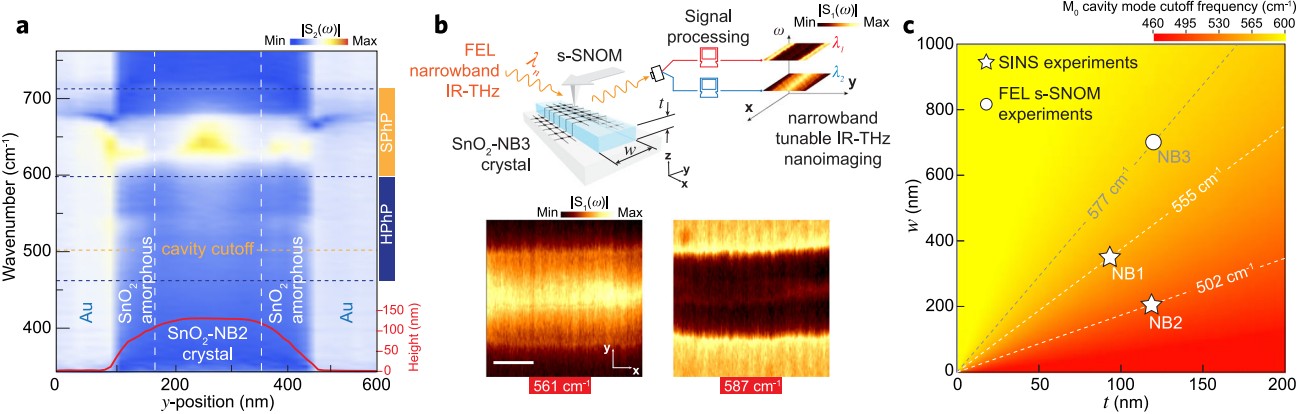

**Fig. 4 SINS and FEL s-SNOM control measurements outside NBs cavity condition. a** Experimental SINS amplitude $S_2(\omega)$ spectral linescan across the $SnO_2$-NB2 crystal ($t = 130$ nm, $w = 200$ nm). Red line profile at the bottom illustrates the AFM topography profile of the NB. Vertical white-dashed lines delimit amorphous and crystalline $SnO_2$ phases. Horizontal blue dark-dashed lines denote the volume (HPhPs) and surface (SPhPs) phonon polaritons spectral ranges. The horizontal orange dashed line indicates the cutoff frequency for this NB. **b** (top) Tunable THz-IR ($\lambda_n$) s-SNOM nano-imaging experimental scheme and FEL s-SNOM narrowband maps (bottom) of the $SnO_2$-NB3 ($t = 117$ nm, $w = 700$ nm) for the frequencies 561 and 587 cm$^{-1}$. Scale bar represents 300 nm. **c** Cavity cutoff frequency map for $M_0$ modes as a function NBs thickness ($t$) and width ($w$). Isofrequency diagonal-dashed lines denote $M_0$ mode cavity cutoff for the $SnO_2$-NB1, -NB2, and -NB3, analyzed by SINS and FEL s-SNOM in this work.

reflection at the edges, and $n$ is the FP resonance order. Due to the tip contribution, the distance between adjacent maxima of the near-field profile is $\lambda_{M0}/2$ for $n > 1$. Therefore, the momentum $q_{M0}$ can be model-extracted by fitting the following equation to the experimental amplitude profiles (Fig. 3c):

$$I = A\sin(2q_{M0}y)F\left(y - \frac{w}{2}, \gamma\right) + B \qquad (5)$$

This approach resembles the case of Alfaro-Mozaz et al.[11], where a similar equation is applied to extract the momentum of longitudinal hybridized surface modes along with a resonating FP hBN antenna. Additionally, we multiply the sinusoidal wave by an effective loss factor, $F(y, \gamma) = \cosh[-\gamma y]$, to take into account an effective damping factor $\gamma$. The model-extracted $q_{M0}$ values, for the selected frequencies, are plotted as white circles superposed to the ($\omega-q$) dispersion relation for the HPhP $M_0$ modes (Fig. 3d). We note a reasonable agreement between theory and the model-extracted $q_{M0}$. Using such values of $q_{M0}$ as inputs into the Eq. 4, we found that $\varphi$ is nearly invariant with $n$, which allows us to calculate the mean reflection phase $<\varphi> = -0.3\pi$. Hence, by inserting $<\varphi>$ in Eq. 4 and considering the lowest FP order $n = 0$, we obtain the relation $q_{M0}w = 1.3\pi$ that, taken to Eq. 3, enables determining the cutoff frequency map of the FP cavity as a function of $w$ and $t$, as plotted in Fig. 4c discussed in detail below.

As a further matter, the FP model provided an effective lifetime estimate for the cavity modes of $\tau_{cv} \approx 0.10 \pm 0.06$ ps, which was obtained from the effective damping factor $\gamma$, a fitting parameter in Eq. 5 that accounts for the intrinsic damping of the $M_0$ mode and radiative losses that occurs upon the reflection of this mode at the NB edges (see Supplementary Note 4 for more details). For a comprehensive assessment of standing waves in $SnO_2$-NBs, we employed nanoscale spectral imaging on the analysis of additional NBs with different morphologies and, therefore, supplementary cavity parameters. Figure 4a shows a SINS spectral linescan of the $SnO_2$-NB2 ($t = 130$ nm and $w = 200$ nm, see AFM red line profile at the bottom) with full coverage of the PhPs frequency ranges. In the SPhPs range, a similar intensity pattern is observed compared to a similar experiment on the $SnO_2$-NB1 (Fig. 2e), confirming that the SPhPs are not influenced by the NBs dimensions studied here. However, in the $RB_1$, no HPhP activity

is observed. This absence of HPhP response can be accurately explained regarding the cutoff frequency, indicated by the horizontal orange dashed line at $502$ cm$^{-1}$, required for the resonant HPhPs to exist in this specific FP cavity. Additionally, Fig. 4b presents free-electron laser far-IR narrowband (FEL s-SNOM) nano-images of a 700 nm wide and 117 nm thick NB (sample $SnO_2$-NB3). The FEL s-SNOM experiment is described in Fig. 4b (top), where the IR-THz tunable FEL illuminates a metallic AFM tip (near-field source/probe) in a self-homodyne detection scheme for the reconstruction of wavelength-selected ($\lambda_n$) narrowband far-IR images (see Methods). Narrowband images at 561 and 587 cm$^{-1}$ (Fig. 4b, bottom) uncover a severe contrast modulation of the near-field response inside the NB, which is explained by the in-between frequency cutoff predicted for this FP cavity (577 cm$^{-1}$).

Figure 4c illustrates a false-color map of $M_0$ cutoff frequencies as a function of the FP cavity form factor. The red-yellow color scale spans the whole $RB_1$ and the cavity dimensions ranges are defined based on the sample geometries analyzed in this work. Isofrequency diagonal-dashed lines, for the FP cavities experimentally approached, indicate the ($w$, $t$) parameters corresponding to $n = 0$. For the nanostructures analyzed by SINS (white star data points), the frequencies 555 cm$^{-1}$ and 502 cm$^{-1}$ indicate the FP cutoffs for the samples $SnO_2$-NB1 and $SnO_2$-NB2, respectively. For the FEL s-SNOM nano-imaging experiment (white circle data point), the map displays a cutoff line at 577 cm$^{-1}$ for a FP cavity similar to the $SnO_2$-NB3, which explains the quasi-negligible near-field response within this NB for an excitation frequency above the FP cutoff (Fig. 4b, imaged at 587 cm$^{-1}$).

In this work, we introduce $SnO_2$ as a nanophotonic material suitable for multimode polaritonics in the mid- to far-IR frequency ranges. The results reported here place $SnO_2$ in the list of natural anisotropic hyperbolic materials for extreme light confinement (e.g., hBN and $MoO_3$) and extend the range of applications of this class of material towards the THz range. As a further step into applications, we studied the cavity confinement properties of as-grown $SnO_2$-NBs exploring their lithography-free advantage. s-SNOM combined with accelerator-based synchrotron and free-electron laser sources enabled far-IR nanoscale spectral imaging of $SnO_2$-NBs, uncovering an HPhP confinement mechanism consistent to resonant cavities. Hence, our

experimental data is interpreted on the basis of a FP model, that consistently explains the measurements. In summary, this work provides a complete nano-optical description of $SnO_2$ as a dielectric and also as a resonator. For a $SnO_2$-NB with $93 \times 350$ $nm^2$ ($t \times w$) cross-section, we present $SnO_2$'s permittivity and PhPs dispersion relation, covering the $RB_1$ up to the SPhPs ranges, supported by experimental confirmation by SINS point spectra and linescan analyses, respectively. Inputting HPhP volume modes ($M_0$) into the FP model allowed the reconstruction of the dispersion relation for the cavity modes $M_0$, which was further attested by SINS hyperspectral analysis. From the model, we estimate a confinement factor of ~50 and an effective lifetime of ~$0.10 \pm 0.06$ ps for the $M_0$ cavity modes. Additionally, we measure two extra NB samples as supplementary evidences for the $SnO_2$ cavity confinement study, including a control experiment using FEL s-SNOM to demonstrate the presence/absence of $M_0$ modes below/above the cutoff frequency. Now confirmed as a hyperbolic medium highly feasible for THz subdiffractional resonators, we foresee $SnO_2$-NBs as an essential building block in modern photonics, opening opportunities for light manipulation in the far-IR range.

## Methods

**Sample preparation**. The $SnO_2$-NBs analyzed in this work were synthesized via gold-catalyst-assisted vapor-liquid-solid (VLS) method[21]. A suitable amount of pure Sn powder (1 g, 4 N pure) was placed on top of a highly p-doped Si substrate coated with a 300-nm-thick amorphous $SiO_2$ layer. A 5 nm Au layer was previously deposited on the Si/SiO$_2$ substrate, to serve as a catalyst. A tube furnace was heated up to 800 °C in an air/argon atmosphere, and the temperature was kept constant for 2 h. Low $O_2$ concentration on the tube furnace atmosphere is important during growth, in order to create shallow-level defects. After a 2 h cooling, $SnO_2$-NBs were found on the Si/SiO$_2$ substrate surface in a cotton-wool-like form. Finally, the $SnO_2$-NBs were removed from the original substrate, ultrasonically dispersed in isopropanol, and then transferred to a fresh thermally evaporated Au(100 nm)/Si surface by drop coating. In this way, isolated NBs can be individually studied as illustrated in Fig. 1. The structure of the as-grown nanostructures was determined by X-ray diffraction (XRD) measurements[34,52]. They can be ascribed to the tetragonal Rutile $SnO_2$ structure (P4$_2$/mnm, space group 136) with lattice constants $a = b = 0.473$ nm, $c = 0.318$ nm, according to the unit cell shown in Fig. 1a.

Scanning electron microscopy (SEM) images of the nanobelts were performed in a FEG Quanta 200 FEI microscope. Transmission electron microscopy (TEM) images of an individual $SnO_2$ nanobelt were performed in a Tecnai G2-20 SuperTwin FEI 200 kV microscope.

**Synchrotron infrared nanospectroscopy (SINS)**. SINS experiments were performed at the Advanced Light Source (ALS)[53] and at the Brazilian Synchrotron Light Laboratory (LNLS)[41,54]. Both beamlines use a quite similar optical setup comprising of an asymmetric Michelson interferometer mounted into a commercial s-SNOM microscope (NeaSnom, Neaspec GmBH), which can be basically described by an AFM microscope possessing a suited optical arrangement to acquire the optical near-field. In the interferometer, the incident synchrotron IR beam is split into two components by a beamsplitter defining the two interferometer arms formed by a metallic AFM tip and an IR high-reflectivity mirror mounted onto a translation stage. The IR beam component of the tip arm is focused by a parabolic mirror on the tip–sample region. In the experiment, the AFM operates in semi-contact (tapping) mode, wherein the tip is electronically driven to oscillate (tapping amplitude of ~100 nm) in its fundamental mechanical frequency $\Omega$ (~250 kHz) in close proximity to the sample surface. The incident light induces an optical polarization to the tip, primarily, caused by charged separation in the metallic coating, the so-called antenna effect. The optically polarized tip interacting with the sample creates a local effective polarization. The back-scattered light stemming from this tip–sample interaction, is combined on the beamsplitter with the IR reference beam from the scanning arm and detected with a high speed IR detector. A lock-in amplifier having $\Omega$ as the reference frequency demodulates the signal and removes the far-field contributions. The resulting interference signal is Fourier-transformed to give the amplitude $\left|S_n(\omega)\right|$ and phase $\Phi_n(\omega)$ spectra of the complex optical $S_n(\omega) = \left|S_n(\omega)\right| e^{i\Phi_n(\omega)}$. All SINS spectra were measured for $n = 2$, i.e., $S_2(\omega)$. For the mid-IR measurements in the LNLS, we used a mercury cadmium telluride detector (MCT, IR Associates) and a ZnSe beamsplitter in the interferometric setup. For the far-IR measurements at ALS Beamline 2.4, a customized Ge:Cu photoconductor, which provides broadband spectral detection down to 320 cm$^{-1}$, and a KRS-5 beamsplitter were employed. The spectral resolution was set as 10 cm$^{-1}$ for a Fourier processing with a zero-filling factor of 4. All spectra in this work were normalized by a reference spectrum acquired on a clean gold surface (100-nm-thick Au sputtered on a silicon substrate).

**Free-electron laser scattering scanning near-field optical microscopy (FEL s-SNOM)**. FEL-s-SNOM was performed at the free-electron laser FELBE at Helmholtz-Zentrum Dresden-Rossendorf (Dresden, Germany). FELBE provides pulsed narrowband radiation (spectral width of ~1% of central wavenumber) at mid-to-far-IR wavenumbers from 40 to 2000 cm$^{-1}$ at a repetition rate of 13 MHz[55]. The IR radiation is focused onto the tip (Pt–Ir-coated Si cantilever) of a home-built s-SNOM setup[42–44] that uses a self-homodyne detection scheme, with the latter leading to a mixed response of optical amplitude and phase[56]. The back-scattered light is detected with a MCT detector. Similar to SINS, demodulation at higher harmonics of the tip-tapping frequency (~160 kHz for our case) is used to differentiate the near-field signal from the far-field background. AFM tapping amplitude was set as ~100 nm peak-to-peak. FEL-s-SNOM 2D $|S_1|$ amplitude scans as shown in Fig. 4b are obtained while keeping all FELBE parameters fixed, i.e., at a fixed wavenumber. The spectral response is obtained by repeating the measurements after tuning the FEL to a different wavenumber.

**Numerical simulations**. The simulation results shown in Fig. 2 were obtained by finite-difference time-domain (FDTD) calculations assisted by the commercial code Lumerical FDTD v8.23. We considered an infinite long rectangular cross-section nanobelt on Au substrate with a thickness ($t$) and width ($w$) of 93 and 350 nm, respectively. The Au dielectric function was taken from Palik[57]. Both axial and transverse components of the $SnO_2$ permittivity, $\varepsilon_{xx}$, $\varepsilon_{yy}$, and $\varepsilon_{zz}$, were described by a Lorentz model (Supplementary Note 1). The metallic tip was approximated by a dipole source located at 300 nm above the crystal surface[11]. In this approach, we assume that the polarizability of the dipole is weakly affected[58] by the polaritons of $SnO_2$. In contrast to usual dipole models for the tip, whose effective dipole moment depends on the exciting field and the polarizability of a sphere[59], the dipole moment of the simulated source is constant. Hence, tip–sample coupling effects are not considered here. The point spectrum presented in Fig. 2d was calculated by probing the normal component of the electric field located under the dipole source for a height ranging from 50 nm below to 200 nm above the crystal–substrate interface. This allows one to probe mainly the evanescent/confined fields that efficiently couples to the tip. For each vertical position, these values were normalized by the simulated normal component of the electric field without the $SnO_2$-NB. The final spectrum was calculated by averaging these normalized values for all probed heights. The field profiles shown in Fig. 2g were calculated by probing the magnitude of the electric field in the cross-sectional region of the $SnO_2$-NB. The field signal was apodized below 400 fs to exclude the dipole source excitation from the frequency-domain data.

Additionally, divergences from the simulated to the experimental spectrum in Fig. 2d can be attributed to (i) divergences from the theoretical and the experimental dielectric tensor, (ii) approximation of the $|S_n(\omega)|$ s-SNOM amplitude to the simulated $|E_Z|$[60], (iii) presence of $SnO_2$ amorphous layer which is not considered in the simulations, (iv) approximation of the topography of the NB to an ideal rectangle with sharp edges, and (v) intrinsic experimental features that cannot be reproduced by this numerical method, such as the shape of the tip, modulation and demodulation of the tip-scattered signal, and tip–sample coupling effects.

## Data availability

The source data that support the findings of this study are available from the corresponding author upon reasonable request. All these data are directly shown in the corresponding figures without further processing.

## Code availability

Scripts for theoretical predictions and contour plot processing are available from the corresponding authors upon reasonable request.

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

## Acknowledgements

We thank the Brazilian Synchrotron Light Laboratory (LNLS), Advanced Light Source (ALS), and Helmholtz-Zentrum Dresden-Rossendorf (HZDR) for providing beamtime for the experiments. Parts of this research were carried out at ELBE at the Helmholtz-Zentrum Dresden-Rossendorf e. V., a member of the Helmholtz Association. I.D.B., E.R. V., and J.C.G. thank Geraldo M. Ribeiro (UFMG) for preliminary studies of production and characterization of tin oxide nanobelts and A. Gobbi (LNNano), and M.H.O. Piazetta (LNNano) for the support on samples preparation. R.O.F. and I.D.B. thank T.M. Santos (LNLS) and A. Cernescu (Neaspec) for technical assistance. I.D.B. and A.M. acknowledge the financial support from the Brazilian Nanocarbon Institute of Science and Technology (INCT/Nanocarbono). F.C.B.M. and F.H.F. acknowledge the CNPq project 140594/2020-5. R.O.F. and R.A.M. acknowledge the FAPESP project 2019/08818-9. R.O.F. acknowledges the support from CNPq through the research grant 311564/2018-6 and FAPESP Young Investigator grant 2019/14017-9. I.D.B. acknowledges the support from CNPq through the research grant 311327/2020-6. J.C.G. acknowledges the financial support of Brazilian agencies CNPq and FAPEMIG. L.W., L.M.E., and S.C.K. acknowledge funding by the BMBF under grants 05K16ODA and 05K19ODB, as well as by the Würzburg-Dresden Cluster of Excellence on Complexity and Topology in Quantum Matter (ct.qmat) and by the TU Dresden graduate academy. E.R.V. would like to acknowledge the CNPq projects 403360/2016-1 and 311534/2017-1, the Center of Microscopy at the Universidade Federal de Minas Gerais (UFMG) and Universidade Tecnológica Federal do Paraná

(CMCM-UTFPR-CT) for providing the equipment and technical support for the electron microscopy experiments. This research used resources of the Advanced Light Source, a U. S. DOE Office of Science User Facility under contract no. DE-AC02-05CH11231.

## Author contributions

I.D.B. together with E.R.V. and J.C.G. initiated the research. E.R.V. and J.C.G. prepared the samples. A.M. provided the interpretation of the crystalline structure of the nanoblets. I.D.B., H.A.B., F.C.B.M., and R.O.F. carried out the SINS experiments. J.M.K., L.M.E., L.W. , and S.C.K. prepared the instrumentation for the FEL measurements. I.D. B., L.W., and F.C.B.M. carried out the FEL experiments. I.D.B., R.A.M., and F.H.F. performed post-experimental data analysis. I.D.B., R.O.F., F.H.F., and R.A.M. interpreted the theoretical approach of polaritonic modes. R.A.M. developed the FDTD simlulations. All authors took part in the interpretation of the phenomena. I.D.B., F.H.F., R.A.M., and R.O.F. prepared the manuscript.

## Competing interests

The authors declare no competing interests.
