## [Peer Review File · Nature Communications]

REVIEWER COMMENTS

Reviewer #1 (Remarks to the Author):

This work establishes SnO₂ nanostructures as an attractive platform for nanophotonics using confined phonon polaritons. Through careful measurements and analysis the authors provide detailed descriptions of bulk and confined modes through SNOM measurements using broadband synchrotron and narrow band FEL sources combined with analytical and numerical modelling. They also image phonon polariton standing waves and extract the mode lifetime.

The paper is well organized and clearly presented. I recommend publication of this interesting work in Nature Communications.

Since the paper stresses on quantitative analysis, it would be useful to include some comments on the following points:

(1) Equations 2 and 3 are derived based on a multi-layer model but applied to a nanoribbon (nanobelt) of width much smaller than the wavelength. A discussion on the applicability of such a model and how well it compares with the numerical calculation that uses the actual geometry is warranted.

(2) How well does the measured M00 mode life time (1.5 ps) compare with what is expected from intrinsic phonon lifetime assumed in the model? Such a comparison could shed light on the quality of the samples and extrinsic sources of mode decay.

Reviewer #2 (Remarks to the Author):
REVIEW OF "SUB-DIFFRACTIONAL WAVEGUIDING OF TERAHERTZ
HYPERBOLIC PHONONS POLARITONS IN TIN OXIDE"

The manuscript by Flavio H. Feres, Rafael A. Mayer *et al* present relevant findings on the optics of tin oxide (SnO₂) at terahertz frequencies. By using accelerator-based IR-THz sources in conjunction with s-SNOM they are able to measure the polaritonic response of SnO₂ nanobelts. They find that inside one of the Reststrahlen bands of SnO₂ (between 430-620 cm⁻¹) the material present a hyperbolic response, being the tensor definite negative for frequencies from 620 to 686 cm⁻¹. The phonon-polaritons of SnO₂ in these frequency ranges possess high lifetimes. As such, Tin Oxide adds to the library of materials presenting deeply confined phonon polaritons, in a region of the THz range that is not covered by other phononic materials such as MoO₃, h-BN or SiC. Thus it would be of great interest for boosting the light absorption in THz photodetectors or for surface-enhanced spectroscopies at these frequency window.

The claims of the paper possess enough novelty, and would be of interest on the field THz nanophotonics and material science. This is due to the fact that, for many technologically relevant frequency ranges in the THz, materials presenting high field confinement, low ohmic losses and the ability to be manufactured in certain geometries are scarce.

The sample preparation and the near-field studies are correctly done. However, the analysis of the near-field data is confusing at times, and at some point, simply incorrect. I would extend myself later. The writing is a little bit confusing at times, and the images could be rearranged in a more explanatory way, as I will discuss in the following. Due to the interesting experimental results, but the lacking explanations and proper analysis, I would invite the authors to revise their manuscript to address specific concerns before a final decision is reached.

The most important flaw is the use of Eq. (4), that I reproduce here for clarity:

$$q_{M0n}(\omega, w) = \sqrt{q_{M0}^2(\omega) - \left[\frac{(n+1)\pi + \phi}{w} \right]^2}, n = 0, 1, 2, 3, \dots \quad (4)$$

This formula approximately gives you the wavevector, q_{M0n} , of the polaritonic modes **along** the nanobelt (as a function of frequency and width). For measuring the wavevector of these modes, you would need to launch the polaritons by the tip. Then the polaritonic mode propagates **along** the nanobelt, gets reflected at one nanobelt end, propagate back to the tip, and finally the tip scatter the fields to the far field. Thus, by raster-scanning the tip on top of the nanobelt you would see modulations with a period which is half of that of the polaritonic mode. See Ref. 1.

However, in this case, the nanobelt present a length that can be considered infinite, thus you are not able to map the M0n wavelengths (that is, measuring near-field signal oscillations along - not across - the nanobelt). Hence, the subsequent analysis is very misleading, and even wrong (with the provided data).

I think that the analysis of modes in ribbons is simpler, and analogous to that carried on in graphene, see Refs. 2 and 3. Applied to this system, you only need to consider that the M0 mode propagating **across** the nanobelt acquire a $2\pi n$ phase in one roundtrip. No extra waveguide modes (M00, M01...), which I emphasize, would be propagating **along** the ribbon, something not measured in the presented manuscript.

Hence, Figure 3a is a wrong comparison.

Also, even being incorrectly compared, there is no enough explanation about the experimental points in Fig. 3a. These are extracted from Fig. 2f, as explained in the figure caption. However, no explanation on how the maxima were identified in Fig. 2f. The signal in Fig. 2f does not provide enough information to determine resonant frequency NF peaks (specially with the spectral resolution shown in Fig. 3a, where different points differ by frequencies comparable to the resolution of the experiment), or the number of maxima across the nanobelt (the colormap shows spatially wide maxima that does not show discernible signal peaks in space).

The problem of calculating modes that propagate along the nanobelt is translated to other portions of the manuscript. Such as Fig. 3b, Figs. 4b and 4e.

Again, when Eq. 5 is used, (which is used to obtain the data points in Fig. 3a. This is misleading, since in the figure caption it is said that is extracted from Fig. 2f) it is claimed that you obtain the wavevector q_{M00} . And again, q_{M00} is the wavevector in the direction **along** the nanobelt (not across, or at an arbitrary angle). Moreover, the fact that the number of maxima steadily increases in Fig. 3d is a clear indication of different FP orders (of the M0 mode) in the ribbon. See Ref. 4 for the case in graphene ribbons (which is conceptually similar).

This is a deep flaw in the paper, which, given the good experimental results, can be overcome with a different analysis.

Other comments:

- By comparing Figures 2d and 2f, which correspond to the same nanobelt, I would say that the colormap of 2f is reversed, since in Fig. 2d there is a maxima at 620cm^{-1} , that, using the legend of Fig. 2f, would be a minima (even at the center of the crystal where the spectra in Fig. 2d is presented).
- In Page 7, change the phrase

Moreover, the branches slope ($\partial\omega/\partial q$) inside the RB1 is negative, which is a similar behavior observed in the RB1 of hBN slabs⁴⁵.

The slope (group velocity) of the Mn modes is positive. The phase velocity is negative (the momentum is negative, thus it should be reflected around the Y axis). See Ref. 5 for a similar experiment.

- In page 11, the following phrase is not correct.

Inside the RB₁, only cavities wider than 200 nm support waveguide modes, which explains the absence of HP2s activity in Figure 4a.

From Fig. 2b: the M1, M2, M3... modes present higher momenta. By introducing these modes to calculate the waveguide modes along the nanobelt, Eq. 4, you would see that for example the M20 mode would probably propagate. Your highlighted phrase refers only to those waveguide modes originating from the M0 mode, and not from high order modes (which are also supported by the nanobelt).

- Figures 2a and 2d are one below to another, however the X axis is different between both, which makes it very misleading. Figure 2d and 2e could be better compared if one was below the other with a similar x-axis. Figure 2g lacks axis. Figure 2d, e, f doesn't have units (even (a.u.)). For the case of the near-field, it would be interesting to present the signal, for example, compared to that of the substrate $|s_2|/|s_{2,Au}|$.
- Why the FEL-s-SNOM scan in Fig. 4d is modulated at the first harmonic? It is due to the intrinsic signal to noise ratio of the homebuilt s-SNOM?

[1] Dolado, I., Alfaro-Mozaz, F. J., Li, P., Nikulina, E., Bylinkin, A., Liu, S., Edgar, J. H., Casanova, F., Hueso, L. E., Alonso-González, P., Vélez, S., Nikitin, A. Y., Hillenbrand, R., *Nanoscale Guiding of Infrared Light with Hyperbolic Volume and Surface Polaritons in van der Waals Material Ribbons*. Adv. Mater. 2020, 32, 1906530. <https://doi.org/10.1002/adma.201906530>

[2] Fei, Z., Rodin, A., Andreev, G. et al. *Gate-tuning of graphene plasmons revealed by infrared nano-imaging*. Nature 487, 82–85 (2012). <https://doi.org/10.1038/nature11253>

[3] Chen, J., Badioli, M., Alonso-González, P. et al. *Optical nano-imaging of gate-tunable graphene plasmons*. Nature 487, 77–81 (2012). <https://doi.org/10.1038/nature11254>

[4] Z. Fei, M. D. Goldflam, J.-S. Wu, S. Dai, M. Wagner, A. S. McLeod, M. K. Liu, K. W. Post, S. Zhu, G. C. A. M. Janssen, M. M. Fogler, and D. N. Basov *Edge and Surface Plasmons in Graphene Nanoribbons*, *Nano Letters* **2015** 15 (12), 8271-8276 DOI: 10.1021/acs.nanolett.5b03834

[5] Yoxall, E., Schnell, M., Nikitin, A. et al. *Direct observation of ultraslow hyperbolic polariton propagation with negative phase velocity*. Nature Photon 9, 674–678 (2015). <https://doi.org/10.1038/nphoton.2015.166>

Reviewer #3 (Remarks to the Author):

In this manuscript, the authors present an experimental demonstration of hyperbolic phonon polaritons in tin oxide nanobelts. Scattering near field optical microscopy is employed to characterize these hyperbolic modes excited via IR and THz radiation generated by a synchrotron and free electron laser. FDTD simulations and analytical calculations are presented to support the experimental findings.

Before commenting on the specifics of the manuscript, I would like to make a general comment on the novelty of this work. Hyperbolic phonon polaritons were first experimentally observed in hexagonal boron nitride (hBN) in 2014 using an s-SNOM (Science, Vol. 343, Issue 6175, pp. 1125-1129, 2014). Earlier this year, there was another published work from Rainer Hillenbrand group (Adv. Mater. 32, 1906530, 2020) which presented an s-SNOM based experimental demonstration of phonon polariton waveguiding in hBN ribbons. The similarity of the current manuscript in the experimental approach as well as the analysis is quite striking particularly with respect to the Adv Mater paper. In this sense, the only novelty I see from this work is more on the materials side (SnO₂). In this context, I should also remark that SnO₂, much like hBN is also a uniaxial hyperbolic material. Thus I do not think the current work presents any fundamental conceptual advance on top of the existing body of research literature on hyperbolic phonon polaritons. Thus while this work is interesting, I think it is suitable for a specialized journal rather than in Nature Communications.

Regarding the specifics of the manuscript, I list below certain points which need further clarification. I hope these points will help the authors improve the manuscript:

1) Regarding Fig. 2(f) and its description in the main text: In the main text, Fig. 2(f) is described as follows: "Within the RB1 range, the near-field intensity fringes produce an interference pattern of frequency-dependent standing waves, suggesting a transverse volume confinement of HP2 waves inside the NB". While the expectation from the physics is correct, the claimed "intensity fringes" are not observable in the region labeled HP2 in the SINS data in Fig. 2(f).

2) Comparing Figures 2(d) and 2(e): where the experiment and simulation results are presented respectively: It is notable that the SPhP resonance frequency appears to be inconsistent between the experiment and the simulation. This needs an explanation.

3) In the caption of Fig. 3(a), the authors mention that they employ a reflection phase of $\pi/2$. This is the same phase as in the hBN case from the Adv Mater paper. Is there a fundamental reason for the phase being identical in the two cases? Why was $\pi/2$ chosen in your case. This question is important because the choice of ϕ affects your cutoff.

4) Regarding the lifetime calculation, my understanding is that the propagation length should be along the longer axis of the waveguide. But on page 11, the authors state that the propagation length is L_y . According to Fig. 3(c) however, 'y' is the transverse dimension. This makes the numbers for the lifetime questionable: this point needs clarification.

5) The authors need to provide some error bars for the lifetime -- say how it varies across samples. How does this measured value compare to theoretical / numerical calculation?

Response

We thank the reviewers for the careful assessment of our work. We have addressed all their comments and believe that these additions have improved our manuscript. Below, we provide a point-by-point reply, with the reviewers' comments in *blue italics* and our response in Black.

Reviewer #1 (Remarks to the Author):

This work establishes SnO₂ nanostructures as an attractive platform for nanophotonics using confined phonon polaritons. Through careful measurements and analysis, the authors provide detailed descriptions of bulk and confined modes through SNOM measurements using broadband synchrotron and narrow band FEL sources combined with analytical and numerical modelling. They also image phonon polariton standing waves and extract the mode lifetime.

The paper is well organized and clearly presented. I recommend publication of this interesting work in Nature Communications.

We appreciate the effort of the reviewer in the assessment of our work and we thank him/her for recommending it for publication in Nature Communications.

Since the paper stresses on quantitative analysis, it would be useful to include some comments on the following points:

(1) Equations 2 and 3 are derived based on a multi-layer model but applied to a nanoribbon (nanobelt) of width much smaller than the wavelength. A discussion on the applicability of such a model and how well it compares with the numerical calculation that uses the actual geometry is warranted.

Using the Eq. 2-3, based on a multi-layer model, we calculated the frequency-momentum dispersion relation for the M_n modes (M_0 is the one experimentally accessed in the work) within the hyperbolic spectral region (Figure 2b-c). It is important to notice that the multi-layer model (Eq. 2-3) is only influenced by the thickness of the SnO₂ slab, since the polariton wavelengths are typically smaller than the studied NB widths.

For the NB, we consider that the M_0 modes become transverse standing waves within the NB (Fabry-Perot cavity), acquiring a round-trip phase when reflected by the edges. Our model is inspired by the approaches in I. Dolado *et al.* Adv. Mat. 2020 and J. Chen *et al.* Nature 2012 for transverse polaritons cavities in other systems. The transverse cavity model allowed extraction of M_0 momentum values (Figure 3d) from experimental data fitting (Figure 3c) that are consistent to the theoretical dispersion relation for this mode attesting the hypothesis of standing waves. FDTD numerical simulations (Figure 2g) corroborate the experimental data as they directly display transverse volume waves confined within the NB. Therefore, our approach can be applied to any system that supports HP² waves confined in a resonant cavity.

We have added a brief discussion on the applicability of the model as well as how the FDTD numerical simulations connect with the model by providing a qualitative and direct visualization of surface/volume polaritons modes within the SnO₂-NBs. Moreover, we stress the complexity of the real SnO₂-NBs crystal structure compared to the ideal geometry defined in the numerical simulations (morphology influence). In page 9, we have modified the phrase to:

“ By comparing experiment and numerical simulation in Figure 2d, a reasonable match is noticed for the central frequency of the main spectral features. In the line shape comparison, there is a fair correspondence between experiment and modeling for the SPhPs (peaks above 600 cm^{-1}), while the HP² range appears to be less trivial to model since volume waves are highly sensitive to the morphology of the NB.”

(2) How well does the measured M00 mode life time (1.5 ps) compare with what is expected from intrinsic phonon lifetime assumed in the model? Such a comparison could shed light on the quality of the samples and extrinsic sources of mode decay.

We have improved our model in order to consider only NB-transverse waves, and therefore, only M₀ modes. Hence, the revised manuscript (please see page 10 for the detailed description of the revised model and/or section S4 in the supplementary material) brings a new estimate for effective lifetime of M₀ modes (0.10 ± 0.06) ps. Additionally, we also calculated the theoretical lifetime 0.3 ps (theoretical predictions of momentum and damping relations) expected for an infinite SnO₂ 2D slab (no cavity), as suggested by the reviewer. As the model-extracted experimental values for the effective lifetime are consistent to the theoretical prediction for a perfect crystal, we suppose our NBs are highly crystalline. Moreover, the crystalline quality of this type of NBs was also previously¹⁻³ confirmed by High Resolution TEM.

In Figure S6, we display the experimental determined lifetimes of the cavity modes, extracted by the fitting described above. According F. J. Alfaro-Mozaz *et al.*⁴, the lifetime can be determined from $\tau_n = Q_n/2\omega_n$, where Q_n is the quality factor of the n-order mode (considered here q/γ)^{5,6}, and ω_n the respective excitation frequency. Figure S6 show a comparison among experimental (black symbols) lifetimes and theoretical predictions is reproduced below.

Figure S6: Comparison among experimental lifetimes (black symbols) and theoretical predictions (blue solid line).

Moreover, we included a new section S6 in the supplementary material, intended to compare theoretical and numerical calculations for the lifetime as suggested by Reviewer 3.

=====
Reviewer #2 (Remarks to the Author):

The manuscript by Ingrid D. Barcelos, Flavio H. Feres, Rafael A. Mayer et al present relevant findings on the optics of tin oxide (SnO₂) at terahertz frequencies. By using accelerator-based IR-THz sources in conjunction with s-SNOM they are able to measure the polaritonic response of SnO₂ nanobelts. They find that inside one of the Reststrahlen bands of SnO₂ (between 430-620 cm⁻¹) the material present a hyperbolic response, being the tensor definite negative for frequencies from 620 to 686 cm⁻¹. The phonon-polaritons of SnO₂ in these frequency ranges possess high lifetimes. As such, Tin Oxide adds to the library of materials presenting deeply confined phonon polaritons, in a region of the THz range that is not covered by other phononic materials such as MoO₃, h-BN or SiC. Thus it would be of great interest for boosting the light absorption in THz photodetectors or for surface-enhanced spectroscopies at this frequency window.

The claims of the paper possess enough novelty, and would be of interest on the field THz nanophotonics and material science. This is due to the fact that, for many technologically relevant frequency ranges in the THz, materials presenting high field confinement, low ohmic losses and the ability to be manufactured in certain geometries are scarce.

We thank the reviewer greatly for the extremely precise assessment on our work. We really appreciate that our points of novelty are clear and that the reviewer agrees about their relevance to the field.

The sample preparation and the near-field studies are correctly done. However, the analysis of the near-field data is confusing at times, and at some point, simply incorrect. I would extend myself later. The writing is a little bit confusing at times, and the images could be rearranged in a more explanatory way, as I will discuss in the following. Due to the interesting experimental results, but the lacking explanations and proper analysis, I would invite the authors to revise their manuscript to address specific concerns before a final decision is reached.

The reviewer is correct and, again, we thank him/her for being extremely thorough with our results and for acknowledging the uniqueness of our experimental dataset. Bellow we address point-by-point responses to all the reviewer's concerns.

*The most important flaw is the use of Eq. (4) from the main text. This formula approximately gives you the wavevector, q_{M0n} , of the polaritonic modes **along** the nanobelt (as a function of frequency and width). For measuring the wavevector of these modes, you would need to launch the polaritons by the tip. Then the polaritonic mode propagates **along** the nanobelt, gets reflected at one nanobelt end, propagate back to the tip, and finally the tip scatter the fields to the far field. Thus, by raster-scanning the tip on top of the nanobelt you would see modulations with a period which is half of that of the polaritonic wavelength. See Ref. 1. However, in this case, the nanobelt present a length that can be considered infinite, thus you are not able to map the $M0n$ wavelengths (that is, measuring near-field signal oscillations along – not across – the nanobelt). Hence, the subsequent analysis is very misleading, and even wrong (with the provided data). I would like to understand the motivation of the authors, since maybe I'm missing something.*

Yes, the reviewer is accurate and, indeed, our model was inappropriate for the phenomenon described/measured. We completely revised our modeling that now takes into

account only M_0 modes confined transversely within the NB. The new model is described below in this answer and the discussion is also reproduced in pages 10-11 of the revised manuscript.

The reviewer is also correct when mentioning that the natural length of the NBs prevents the access to possible longitudinal cavity modes. However, we did try to measure the polaritonic oscillations near the NB terminal edge, as suggested, in order to investigate how these longitudinal modes interact with the transverse standing waves. The new data are presented below:

Figure R1. SINS amplitude narrowband maps reconstructed from a full hyperspectral image of SnO₂-NB1/Au for the frequencies 546, 551, 556, 559, 561 and 564 cm⁻¹. Black-dashed lines delimit crystalline and amorphous phases of the NB. Blue arrows guide the eyes highlighting the interference fringes HP² longitudinal waves inside the SnO₂-NB1 crystal. Scale bar represents 100 nm.

The polaritonic images in Figure R1 were obtained from a hyperspectral (HS) map on the end of a SnO₂ nanobelt recently measured at the Advanced Light Source (ALS/Berkeley). The HS slices unveil changes in the longitudinal intensity pattern as we select different wavelengths from the 546 to 564 cm⁻¹. This is an indication of existence of the M_{0n} modes may be interacting with the transverse modes. Even though the Figure R1 shows a clear shift of the longitudinal maxima as we change the wavelength, the spatial resolution of the data does not allow extracting fundamental properties of these longitudinal modes. Because of the COVID-19 pandemic, we got limited beamtime at ALS, thus, we could not circumvent experimental issues to acquire a HS map over a larger area so that we would extract the profile of such longitudinal mode. However, these new data and discussion on the potential M_{0n} modes is now a good motivation for further studies on engineered SnO₂ THz nanowaveguides and resonators. However, further measurements are needed and beyond the scope of this current manuscript.

As a preliminary conclusion, SnO₂ NBs most likely support M_{0n} modes, however, we will limit our discussion on the nanobelt transverse modes observed and will apply a proper model for the phenomenon description, as recommended by the reviewer.

In my opinion, the appropriate analysis would be analogous to that carried on in graphene, see Refs. 2 and 3. Applied to this system, you only need to consider that the M_0 mode propagating across the nanobelt acquire a $2\pi n$ phase in one roundtrip. No extra waveguide modes (M_{00} , M_{01} ...), which I emphasize, would be propagating along the ribbon, something not measured in the presented manuscript.

Indeed, an analysis taking into account only the transverse M_0 modes was definitely more appropriate for our case. Based on the reviewer's suggestions, we are now using a model analogous to the referred works (specially I. Dolado *et al.* **Adv. Mat.** 2020 and J. Chen *et al.* **Nature** 2012) in which we describe the THz polaritonic response of the NBs in the scope of NB-transverse waves in a Fabry-Perot (FP) cavity, as illustrated below:

In this model, the mode M_0 accumulates a round-trip phase (left part of the Eq. 1) that is equal to a multiple integer of 2π (right part of the Eq. 1) to satisfy the FP condition, as depicted in I. Dolado *et al.* for the transverse modes of a hBN waveguide. Hence, the momentum of the M_0 mode, q_{M_0} , must satisfy the following equation:

$$2q_{M_0}w + 2\varphi = (n + 1) 2\pi, \quad (1)$$

where w is the width of the nanobelt, φ is the phase acquired in the reflection at the borders, and n is the FP resonance order. Due to the tip contribution, the distance between adjacent maxima of the near-field profile is $\lambda_{M_0}/2$. Therefore, the momentum q_{M_0} can be model-extracted by fitting the experimental data to a sinusoidal wave, as the following equation:

$$I = A \sin(2q_{M_0}x)F\left(x - \frac{w}{2}, \gamma\right) + B. \quad (2)$$

This approach resembles the case of F. J. Alfaro-Mozaz *et al.*⁴, where a similar equation is applied to extract the momentum of longitudinal hybridized surface modes along a resonating FP hBN antenna. Additionally, we multiply the sinusoidal wave by an effective loss factor, $F(y, \gamma) = \cosh[-\gamma y]$, to take into account an effective damping factor γ . The model-extracted q_{M_0} values, for the selected frequencies, are plotted as white circles superposed to the $(\omega$ - q) dispersion relation for the $HP^2 M_0$ modes (Figure 3d). We note a reasonable agreement between theory and the model-extracted q_{M_0} s. Using such values of q_{M_0} as inputs into the Eq. 1, we found that φ is nearly invariant with n , which allows us to calculate the mean reflection phase of $\langle \varphi \rangle = -0.3\pi$. Hence, substituting $\langle \varphi \rangle$ in the Eq. 1, considering the lowest FP order $n = 0$, we obtain the relation $q_{M_0}w = 1.3\pi$ that, taken to the Eq. 3 (main text), enables determining the cutoff frequency map of the FP cavity as a function of w and t , as plotted in Figure 4c in the main text.

Hence, Figure 3a is a wrong comparison. Also, even being incorrectly compared, there is not enough explanation about the experimental points in Fig. 3a. These are extracted from Fig. 2f, as explained in the figure caption. However, no explanation on how the maxima were identified in Fig. 2f. The signal in Fig. 2f does not provide enough information to determine resonant

frequency NF peaks (specially with the spectral resolution shown in Fig. 3a, where different points differ by frequencies comparable to the resolution of the experiment), or the number of maxima across the nanobelt (the colormap shows spatially wide maxima that does not show discernible signal peaks in space).

The reviewer is correct, and we apologize for the mistaken comparison and incomplete description on the experimental points. The manuscript's Figure 3c is now rearranged as presented below. The experimental profiles were indeed extracted from the spectral linescan and we modified the figure that now overlaps the amplitude profiles and false color spectral linescan (Figure 3c). Horizontal red arrows indicate the frequencies of the extracted profiles. Despite the fact that we have a nominal spectral resolution of 10 cm^{-1} for the spectral linescan, a zero-filling factor of 4 was used in the Fourier processing of the smoothly broadband interferograms, allowing the technique to resolve features beyond the nominal spectral resolution, as confirmed in our experimental lineshape analysis. Below we present the revised Figure 3c.

The problem of calculating modes that propagate along the nanobelt is translated to other portions of the manuscript. Such as Fig. 3b, Figs. 4b and 4e.

Figure 3b and 4b were removed and Figure 4e (now 4c) was updated accordingly to the corrected model.

Again, when Eq. 5 is used, (which is used to obtain the data points in Fig. 3a. Which I dont understand, since in the figure caption it is said that is extracted from Fig. 2f) it is claimed that you obtain the wavevector q_{M00} . And again, q_{M00} is the wavevector in the direction along the nanobelt (not across, or at an arbitrary angle). Moreover, the fact that the number of maxima steadily increases in Fig. 3d is a clear indication of different FP orders (of the M_0 mode) in the ribbon. See Ref. 4 for the case in graphene ribbons (which is conceptually similar).

The reviewer is correct and this problem also appears as a consequence of the inappropriate model we originally suggested. Now, Eq. 5 is only applied to M_0 modes across the NB, as a classic NB-transverse Fabry-Perot cavity.

This is a deep flaw in the paper, which, given the good experimental results, can be overcome with a different analysis.

We really appreciate the reviewer's points that gave us the opportunity to correct and improve our work in its essence. We also thank the reviewer for emphasizing the good quality and uniqueness of our experimental data.

Other comments:

By comparing Figures 2d and 2f, which correspond to the same nanobelt, I would say that the colormap of 2f is reversed, since in Fig. 2d there is a maxima at 620cm^{-1} , that, using the legend of Fig. 2f, would be a minima (even at the center of the crystal where the spectra in Fig. 2d is presented).

We corrected the color scale of Figure 2f (now 2e) that was indeed reversed.

In Page 7, change the phrase:

"Moreover, the branches slope ($\partial\omega/\partial q$) inside the RB1 is negative, which is a similar behavior observed in the RB1 of hBN slabs⁴⁵."

The slope (group velocity) of the Mn modes is positive. The phase velocity is negative (the momentum is negative, thus it should be reflected around the Y axis). See Ref. 5 for a similar experiment.

The reviewer is correct. As we are no longer addressing group velocity in this work, we readjust the sentence in order to avoid misleading interpretations (see Page 7).

In page 11, the following phrase is not correct.

"Inside the RB1, only cavities wider than 200 nm support waveguide modes, which explains the absence of HP2s activity in Figure 4a."

From Fig. 2b: the M1, M2, M3... modes present higher momenta. By introducing these modes to calculate the waveguide modes along the nanobelt, Eq. 4, you would see that for example the M20 mode would probably propagate. Your highlighted phrase refers only to those waveguide modes originating from the M0 mode, and not from high order modes (which are also supported by the nanobelt).

We no longer discuss this point as we removed the original Figure 4b.

Figures 2a and 2d are one below to another, however the X axis is different between both, which makes it very misleading. Figure 2d and 2e could be better compared if one was below the other with a similar x-axis. Figure 2g lacks axis. Figure 2d, e, f doesn't have units (even (a.u.)). For the case of the near-field, it would be interesting to present the signal strength compared, for example, to that of the substrate $|s_2|/|s_{2,Au}|$.

Fig. 2 was rearranged and improved following the suggestions, as presented below:

It would be instructive to explain why the FEL-s-SNOM scan in Fig. 4d is modulated at only the first harmonic. It is due to the intrinsic signal to noise ratio of the homebuilt s-SNOM?

We thank the reviewer for pointing this out. Indeed, the motivation for using the 1st harmonic is the improved SNR in comparison to higher demodulations frequencies. This choice is based on the qualitative similarity between $|S_1|$ and $|S_2|$ contrasts (except by the noise floor) when comparing frequencies below and above the cavity cutoff frequency 577 cm^{-1} . We ascribe this behaviour to the weak influence of the far-field background in the measurement as the illumination wavelengths are considerably larger (up to $20 \mu\text{m}$) than the AFM tapping amplitude ($\sim 100 \text{ nm}$).

In Fig. 4b of the manuscript we show the optical near-field signal of the nanobelt at 561 cm^{-1} and 587 cm^{-1} . Figure R2 reprints these scans and shows two additional scans at adjacent wavenumbers (551 cm^{-1} and 607 cm^{-1}), illustrating the nanobelt behaviour in a broader spectral range. In Fig. 4d and in the top row of Figure R4, we show the near-field signal $|S_1(\omega)|$ using demodulation of the optical signal at the first harmonic. In the bottom row of Figure R2, the same scans are displayed for demodulation at the second harmonic ($|S_2(\omega)|$), verifying that the qualitative behaviour of the nanobelts – particularly the observed cutoff frequency – is independent of the demodulation order. This discussion was added to the Supplementary Information.

Figure R4: Narrowband FEL-s-SNOM scans at four selected wavenumbers close to the cutoff frequency 577 cm^{-1} . Top row: Optical signal $|S_1(\omega)|$ demodulated at the first harmonic of the cantilever frequency. Bottom row: $|S_2(\omega)|$ demodulated at the second harmonic. The scale bar in the top left scan represents 300 nm .

[1] Dolado, I., Alfaro-Mozaz, F. J., Li, P., Nikulina, E., Bylinkin, A., Liu, S., Edgar, J. H., Casanova, F., Hueso, L. E., Alonso-González, P., Vélez, S., Nikitin, A. Y., Hillenbrand, R., *Nanoscale Guiding of Infrared Light with Hyperbolic Volume and Surface Polaritons in van der Waals Material Ribbons. Adv. Mater.* 2020, 32, 1906530. <https://doi.org/10.1002/adma.201906530>

[2] Fei, Z., Rodin, A., Andreev, G. et al. *Gate-tuning of graphene plasmons revealed by infrared nano-imaging. Nature* 487, 82–85 (2012). <https://doi.org/10.1038/nature11253>

[3] Chen, J., Badioli, M., Alonso-González, P. et al. *Optical nano-imaging of gate-tunable graphene plasmons. Nature* 487, 77–81 (2012). <https://doi.org/10.1038/nature11254>

[4] Z. Fei, M. D. Goldflam, J.-S. Wu, S. Dai, M. Wagner, A. S. McLeod, M. K. Liu, K. W. Post, S. Zhu, G. C. A. M. Janssen, M. M. Fogler, and D. N. Basov *Edge and Surface Plasmons in Graphene Nanoribbons, Nano Letters* 2015 15 (12), 8271-8276 DOI: 10.1021/acs.nanolett.5b03834

[5] Yoxall, E., Schnell, M., Nikitin, A. et al. *Direct observation of ultraslow hyperbolic polariton propagation with negative phase velocity. Nature Photon* 9, 674–678 (2015). <https://doi.org/10.1038/nphoton.2015.166>

=====
Reviewer #3 (Remarks to the Author):

In this manuscript, the authors present an experimental demonstration of hyperbolic phonon polaritons in tin oxide nanobelts. Scattering near field optical microscopy is employed to characterize these hyperbolic modes excited via IR and THz radiation generated by a synchrotron and free electron laser. FDTD simulations and analytical calculations are presented to support the experimental findings.

Before commenting on the specifics of the manuscript, I would like to make a general comment on the novelty of this work. Hyperbolic phonon polaritons were first experimentally observed in hexagonal boron nitride (hBN) in 2014 using an s-SNOM (Science, Vol. 343, Issue 6175, pp. 1125-1129, 2014). Earlier this year, there was another published work from Rainer Hillenbrand group (Adv. Mater. 32, 1906530, 2020) which presented an s-SNOM based experimental demonstration of phonon polariton waveguiding in hBN ribbons. The similarity of the current manuscript in the experimental approach as well as the analysis is quite striking particularly with respect to the Adv Mater paper. In this sense, the only novelty I see from this work is more on the materials side (SnO₂). In this context, I should also remark that SnO₂, much like hBN is also a uniaxial hyperbolic material. Thus I do not think the current work presents any fundamental conceptual advance on top of the existing body of research literature on hyperbolic phonon polaritons. Thus while this work is interesting, I think it is suitable for a specialized journal rather than in Nature Communications.

We appreciate the rigor of the reviewer's assessment. We agree that hBN is a highly explored material and that our analytical approach is inspired in pioneer works in the theme. However, we do believe that the extension of the applicability of these polaritonic platforms to unexplored wavelengths is a major breakthrough. SnO₂ nanobelts not only provide a new nano-optical media for transport of highly confined THz waves but also are lithography-free 1D structures that enable cavity confinement of those waves. Our perception is in phase with

the editorial board of Nature Communications that considered those innovations adequate to the impact level and scope of the journal.

Regarding the specifics of the manuscript, I list below certain points which need further clarification. I hope these points will help the authors improve the manuscript:

1) Regarding Fig. 2(f) and its description in the main text: In the main text, Fig. 2(f) is described as follows: "Within the RB1 range, the near-field intensity fringes produce an interference pattern of frequency-dependent standing waves, suggesting a transverse volume confinement of HP2 waves inside the NB". While the expectation from the physics is correct, the claimed "intensity fringes" are not observable in the region labeled HP2 in the SINS data in Fig. 2(f).

We agree with the reviewer that these profiles were not readily noticeable from the contrast of the spectral linescan presented in original Fig. 2f. In order to make this clear to the reader, we organized a new figure (Figure 3c) in which we overlap extracted amplitude profiles and a spatio-spectral false color map (linescan) from where the profiles were extracted. The contrast is now more intuitive and red arrows indicate the frequencies of the extracted profiles.

2) Comparing Figures 2(d) and 2(e): where the experiment and simulation results are presented respectively: It is notable that the SPhP resonance frequency appears to be inconsistent between the experiment and the simulation. This needs an explanation.

It is important to notice that this range of frequency is related to HP₂s and not the SPhPs. The discrepancy between experiment and simulation in the original Figures 2d-e (now Figure 2d) comes from the intrinsic structural complexity of the real NBs, which is highly challenging to be modeled in the simulation. The measured NBs are naturally lithography-free self-assembled structures that are covered by a layer of amorphous SnO₂. Therefore, it is extremely challenging to perfectly control the shape and quality of the NB boundaries in their fabrication. The numerical simulation considers an ideal rectangular solid with perfectly flat surfaces and enables a qualitative view of the local spectral response of the SnO₂-NB, which is rather consistent to the measured spectrum. Moreover, the FDTD simulations consider permittivities calculated/measured for bulk SnO₂, which is not ideal for our 1D structures.

We have added the following sentence in the manuscript (see Page 9) regarding this point:

"By comparing experiment and numerical simulation in Figure 2d, a reasonable match is noticed for the central frequency of the main spectral features. In the line shape comparison, there is a fair correspondence between experiment and modeling for the SPhPs (peaks above 600 cm⁻¹), while the HP² range appears to be less trivial to model since volume waves are highly sensitive to the morphology of the NB."

3) In the caption of Fig. 3(a), the authors mention that they employ a reflection phase of $\pi/2$. This is the same phase as in the hBN case from the Adv Mater paper. Is there a fundamental reason for the phase being identical in the two cases? Why was $\pi/2$ chosen in your case. This question is important because the choice of ϕ affects your cutoff.

The reviewer is correct to question the origins of the reflection phase value. We added corrections to the model, and now the reflection phase is calculated from the model-extracted momentum from the profiles fittings, as described in the main text (page11). We would like to point out that the mean reflection phase $\langle \varphi \rangle = -0.3 \pi$ calculated in this work is the result of complex interactions at the border of the crystal, which arise to satisfy

the continuity of electric and magnetic fields. Thus, non-trivial reflection phase can be found, as discussed in A. Yu. Nikitin et al⁷, where they found a phase shift of $-3\pi/4$ for graphene plasmons resonators.”

4) Regarding the lifetime calculation, my understanding is that the propagation length should be along the longer axis of the waveguide. But on page 11, the authors state that the propagation length is L_y . According to Fig. 3(c) however, ‘y’ is the transverse dimension. This makes the numbers for the lifetime questionable: this point needs clarification.

The reviewer is correct and precise in this questioning. In fact, our originally suggested model was inappropriate. We now limit our analysis to the M_0 modes, i. e., to the transverse waves to the NB (y-axis). So, the polariton lifetime is now estimated from the damping factor calculated for the transverse standing waves (0.10 ± 0.06 ps). The manuscript text was improved (see pages 10-11), and new a section was added to the SI in order to make the model clearer to the readership.

In Figure S6, we display the experimental determined lifetimes of the cavity modes, extracted by the fitting described above. According F. J. Alfaro-Mozaz *et al.*⁴, the lifetime can be determined from $\tau_n = Q_n / 2\omega_n$, where Q_n is the quality factor of the n-order mode (considered here q/γ)^{5,6}, and ω_n the respective excitation frequency. Figure S6 show a comparison among experimental (black symbols) lifetimes and theoretical predictions is reproduced below.

Figure S6: Comparison among experimental lifetimes (black symbols) and theoretical predictions (blue solid line).

5) The authors need to provide some error bars for the lifetime -- say how it varies across samples. How does this measured value compare to theoretical / numerical calculation?

We have included the standard deviation for the estimated lifetime as suggested in a new section in the supplementary information.

We performed FDTD simulations to extract the dispersion relation of M_0 modes in an ideal 93 nm thick SnO_2 slab. As schemed in Figure S5a, the tip was modelled as a moving dipole source located at 300 nm above the gold surface, which scans the sample below⁸. The $|E_z|_{\text{SnO}_2}(z)$ fields were measured below the dipole by a line monitor, which crosses the entire cross-section of the slab. The resulting signal for each position in x direction, showed in Figure S5b, was obtained by integrating along the z direction, the $|E_z|_{\text{SnO}_2}(z)$ normalized by the fields simulated on Au, $|E_z|_{\text{Au}}(z)$.

The simulated linescan shows two distinctive bands which agrees with the predicted surface (ω : 650-700 cm^{-1}) and volume (ω : 465-605 cm^{-1}) hyperbolic polaritons modes in SnO_2 . Moreover, Figure S5c shows the extracted profiles for some selected frequencies. The profiles were fitted by $\frac{A}{\sqrt{2x}}e^{-2x\gamma}\sin(2qx + \varphi)$, that accounts for the propagation of polaritons, with momentum q and damping γ , that propagates with circular decay and are back reflected by the borders⁹. The extracted frequency-dependent q and γ , are presented in Figure S5d, and shows excellent agreement with the predicted M_0 modes. Furthermore, we were also able to calculate the lifetime of M_0 modes, which agrees well with the theoretical values (more on the calculation of the polariton lifetimes can be found in S6 of the Supplementary Information).

Figure S5: FDTD simulations | a) Scheme of the system used to simulate the spectral linescan of an ideal 93-nm-thick SnO_2 slab in the proximities of a sharp border, represented in b). The dipole is represented by the red arrow, while the dashed red line represents the field monitor. c) Extracted line profiles for selected frequencies of (b). d) Model-extracted momentum q , damping γ , and lifetime of M_0 polaritons in SnO_2 . The simulated data are represented by black circles with an associated error represented by the grey bars. The dashed red line represents the theoretical values obtained from Equation 3 in the main text.

References

1. Viana, E. R., Ribeiro, G. M., de Oliveira, A. G. & González, J. C. Metal-to-insulator transition induced by UV illumination in a single SnO₂ nanobelt. *Nanotechnology* **28**, 445703 (2017).
2. Viana, E. R., González, J. C., Ribeiro, G. M. & Oliveira, A. G. d. 3D hopping conduction in SnO₂ nanobelts. *Phys. Status Solidi - Rapid Res. Lett.* **6**, 262–264 (2012).
3. Viana, E. R., González, J. C., Ribeiro, G. M. & de Oliveira, A. G. Electrical observation of sub-band formation in SnO₂ nanobelts. *Nanoscale* **5**, 6439–6444 (2013).
4. Alfaro-Mozaz, F. J. *et al.* Nanoimaging of resonating hyperbolic polaritons in linear boron nitride antennas. *Nat. Commun.* **8**, (2017).
5. Caldwell, J. D. *et al.* Sub-diffractive volume-confined polaritons in the natural hyperbolic material hexagonal boron nitride. *Nat. Commun.* **5**, 5221 (2014).
6. Basov, D. N., Fogler, M. M. & Garcia de Abajo, F. J. Polaritons in van der Waals materials. *Science (80-.)*. **354**, aag1992–aag1992 (2016).
7. Nikitin, A. Y., Low, T. & Martin-Moreno, L. Anomalous reflection phase of graphene plasmons and its influence on resonators. *Phys. Rev. B* **90**, 41407 (2014).
8. Nikitin, A. Y. *et al.* Real-space mapping of tailored sheet and edge plasmons in graphene nanoresonators. *Nat. Photonics* **10**, 239–243 (2016).
9. Woessner, A. *et al.* Highly confined low-loss plasmons in graphene–boron nitride heterostructures. *Nat. Mater.* **14**, 421–425 (2015).

REVIEWER COMMENTS

Reviewer #1 (Remarks to the Author):

The authors have addressed my concerns fully.

Reviewer #2 (Remarks to the Author):

The authors have addressed most of my concerns in the revised manuscript and is now clearer and the interpretation is more robust. However, the current version still raises some concerns that needs a revision:

-Figure 2 show that $\text{Re}(\epsilon_{zz}) < 0$. However in page 6 of the main text is stated that: *“Specifically, the hyperbolic window (shaded in light blue in Figure 2a), defined here as RB type I (RB1) with $\text{Re}[\epsilon_x, y] < 0$, and $\text{Re}[\epsilon_z] > 0$, is delimited in the frequency range 465-605 cm^{-1} . In this case, these modes only exist inside the crystal volume and propagate with a well-known...”*. Moreover, the more usual convention is to denominate the class of polaritons sustained in the Type 1 Reststrahlen Bands as HP^1 and not HP^2 .

-The discussion in Page 6 is unclear. The second paragraph (starting with *“These phonon modes give rise to the RB, ...”*) describes the hyperbolic modes propagating in bulk SnO₂. However, the next paragraph (*“Figure 2b,c display the calculated frequency-momentum (ω - q) PhPs dispersion relation...”*) discusses the modes in a thin SnO₂ layer, different from the previous paragraph, without clearly delimiting that is a different system. Fig. 2b and its inset lacks this distinction (bulk vs thin film) also.

-Modelling the tip sample interaction is not an easy task. Since the authors are modelling it by using a fixed dipole and taking the fields in a line below the dipole, it would be of interest to discuss the potential limitations of the model. For example, the tip modulation and the demodulation of the tip-scattered signal is not taken into account. Neither the potential tip-sample coupling effects. Nor the transition from the near-fields that are simulated and the scattered EM far-fields detected in the experiment. Highlighting the limitations in the model would be informative and helpful to explain potential experiment-simulation disagreements, such as Fig. 2d.

-The discussion of the lifetime is a bit unclear. The experiment don't measure the lifetime of the free-propagating M₀ mode. Instead, the lifetime of the M₀ mode resonating between the two sides of the nanobelt is measured. Thus, the discussion of the lifetime has to take into account in the main text. Does the FP model of the nanobelt, section S6, take into account potential radiative losses in the reflection of the M₀ mode at the edges?

-Regarding the near-field experiments, is the near-field signal normalized to that of any reference material? I don't find it in the methods. The normalization to a reference sample is important since: the laser power can vary from wavelength to wavelength, also the its phase, the detector sensitivity or even the radiation pattern of the probing tip can change from wavelength to wavelength.

-There are several typos along the manuscript, I point out here a couple, but there are a few more. I find a little more polishing necessary.

Page 3: "Lithiation electrode" should be plural.

Page 4: "Hyperbolic behaviours" is a strange choice of words. "FDTD simulation" should be plural

Reviewer #3 (Remarks to the Author):

The authors have addressed the technical objections I raised and with the extensive revisions, the manuscript looks in much better shape now. In particular, the theoretical analysis and their interpretation of experimental results now appear to be valid.

As I alluded to in my first report, I am still not convinced that the central idea of this work -- studying a new uniaxial hyperbolic material -- presents any substantial conceptual advance to merit publication in Nature Comm. Besides there is a significant overlap in the analysis as well as the experiment compared to other papers published this year (eg. Adv. Mater. 32, 1906530, 2020) although the material system studied there is hBN instead of SnO₂. Having said that, I understand that this judgement is somewhat subjective and will therefore leave it up to the editor to take a call on this point.

In summary, the manuscript now appears to be technically sound enough to be published.

Response

We thank the reviewers for the careful assessment of our work. We have addressed all their comments and believe that these additions have improved our manuscript. Below, we provide a point-by-point reply, with the reviewers' comments in *blue italics* and our response in Black.

=====
Reviewer #2 (Remarks to the Author):

The authors have addressed most of my concerns in the revised manuscript and is now clearer and the interpretation is more robust. However, the current version still raises some concerns that needs a revision:

We thank the reviewer for confirming that the manuscript is now more solid. We also thank the reviewer for the precise evaluation of the revised manuscript and for giving us the chance to address additional raised concerns.

-Figure 2 show that $Re(\epsilon_{zz}) < 0$. However in page 6 of the main text is stated that: "Specifically, the hyperbolic window (shaded in light blue in Figure 2a), defined here as RB type I (RB₁) with $Re[\epsilon_{x,y}] < 0$, and $Re[\epsilon_z] > 0$, is delimited in the frequency range 465-605 cm⁻¹. In this case, these modes only exist inside the crystal volume and propagate with a well-known...". Moreover, the more usual convention is to denominate the class of polaritons sustained in the Type 1 Reststrahlen Bands as HP1 and not HP2.

We apologize the confusion and we have corrected this part of the sentence to:

... as RB type I (RB₁) with $Re[\epsilon_{yy}] > 0$, and $Re[\epsilon_{zz}] < 0$, is delimited...

Also, we agree that our terminology can potentially cause some misleading interpretations. Therefore, we are switching the term HP² to HPhP for the Hyperbolic Phonons Polaritons within the Reststrahlen band type I.

-The discussion in Page 6 is unclear. The second paragraph (starting with "These phonon modes give rise to the RB, ..." describes the hyperbolic modes propagating in bulk SnO₂. However, the next paragraph ("Figure 2b,c display the calculated frequency-momentum (ω -q) PhPs dispersion relation...") discusses the modes in a thin SnO₂ layer, different from the previous paragraph, without clearly delimiting that is a different system. Fig. 2b and its inset lacks this distinction (bulk vs thin film) also.

We have improved this discussion as suggested. In fact, the first referred paragraph describes the class of material in a general perspective. We then added a sentence emphasizing that our approach for the theoretical dispersion considers thin films and, therefore, ultra-confined waves:

"Although this is a general description for the SnO₂, our approach is limited to thin films as we explore only ultra-confined phenomena ($q \gg k_0$)."

Complementary in the text of page 6, the original text already explained the approximation we applied in the expression of the extraordinary momentum which is used in the calculation of the ω - q dispersion:

“For the hyperbolic modes, considering highly confined subdiffractive waves ($q \gg k_0$), we can rewrite the z-axis and extraordinary momentum as $k_i = ik_y$ and $k_{ez} = i \sqrt{\frac{\epsilon_{yy}}{\epsilon_{zz}}}$,”

Finally, we have added the thickness ($t = 93$ nm) of the 2D SnO₂ flake considered in the calculation presented in Fig. 2b-c:

“Error! Reference source not found.b,c display the calculated frequency-momentum (ω - q) PhPs dispersion relation for a 2D SnO₂ flake ($t = 93$ nm).”

This information was also included in the Fig. 2b-c.

-Modelling the tip sample interaction is not an easy task. Since the authors are modelling it by using a fixed dipole and taking the fields in a line below the dipole, it would be of interest to discuss the potential limitations of the model. For example, the tip modulation and the demodulation of the tip-scattered signal is not taken into account. Nor the transition from the near-fields that are simulated and the scattered EM far-fields detected in the experiment. Neither the potential tip-sample coupling effects. Highlighting the limitations in the model would be informative and helpful to explain potential experiment-simulation disagreements, such as Fig. 2d.

We appreciate the reviewer for raising this point, which we address in the “Numerical Simulations” Methods section in the main text. Indeed, the simulation of accurate s-SNOM amplitude and phase response is a difficult task due to the non-trivial near-field interaction between the tip and the sample¹.

To highlight the limitations of the numerical modeling, as suggested by the reviewer, we added the following sentences in the main text:

Page 9:

“The divergence between simulated and measured spectrum in Figure 2d can be attributed to realistic experimental aspects not taken into account in the simulation (see Methods).”

Methods:

“In this approach, we assume that the polarizability of the dipole is weakly affected by the polaritons of SnO₂². In contrast to usual dipoles models for the tip, whose effective dipole moment depends on the exciting field and the polarizability of a sphere³, the dipole moment of the simulated source is constant. Hence, tip-sample coupling effects are not considered here.”

... the dipole source for a height ranging from 50 nm below to 200 nm above the crystal-substrate interface. *“This allows one to probe mainly the evanescent/confined fields that efficiently couples to the tip.”* For each vertical position, these values were normalized ...

“Additionally, divergences from the simulated to the experimental spectrum in Figure 2d can be attributed to (i) divergences from the theoretical and the experimental dielectric tensor, (ii) approximation of the $|S_2(\omega)|$ s-SNOM amplitude to the simulated $|E_z|^4$, (iii) presence of SnO₂ amorphous layer which is not considered in the simulations, (iv) approximation of the topography of the NB to an ideal rectangle with sharp edges, and (v) intrinsic experimental features that cannot be reproduced by this numerical method, such as the shape of the tip, modulation and demodulation of the tip-scattered signal, and tip-sample coupling effects.”

-The discussion of the lifetime is a bit unclear. The experiment don't measure the lifetime of the free-propagating MO mode. Instead, the lifetime of the MO mode resonating between the two sides of the nanobelt is measured. Thus, the discussion of the lifetime has to take into account in the main text. Does the FP model of the nanobelt, section S6, take into account potential radiative losses in the reflection of the MO mode at the edges?

We incorporated your observation in the main text (page 11) and described further the effective lifetime in the Supplementary Material. The effective lifetime takes into account all the possible losses that the M₀ mode may undergoes, such as due to the intrinsic damping factors and, as the reviewer mentioned, radiative losses that occur upon the reflection of the mode at the NB edges. As presented in Figure S6 (Supplementary Material), the predicted lifetime is twice the value of the calculated effective lifetime, which is reasonable since the theoretical prediction only considers the intrinsic damping of the M₀ mode.

-Regarding the near-field experiments, is the near-field signal normalized to that of any reference material? I don't find it in the methods. The normalization to a reference sample is important since: the laser power can vary from wavelength to wavelength, also the its phase, the detector sensitivity or even the radiation pattern of the probing tip can change from wavelength to wavelength.

We have added the following missing information in the manuscript's text (page 8) and Methods section, respectively:

“...Each point over the line carries a normalized amplitude ($|S_2| = |S_{SnO_2}| / |S_{Au}|$) spectrum, ...”

“All spectra in this work were normalized by a reference spectrum acquired on a clean gold surface (100 nm thick Au sputtered on a silicon substrate).”

-There are several typos along the manuscript, I point out here a couple, but there are a few more. I find a little more polishing necessary.

Page 3: "Lithiation electrode" should be plural.

Corrected to "Lithiation electrodes"

Page 4: "Hyperbolic behaviours" is a strange choice of words. "FDTD simulation" should be plural

Indeed, the phrase in page 4 was a bit confusing. We rephrased as:

"The inversion of signs of the real parts of the permittivity components in different RBs inside the mid- and far-IR ranges configures the hyperbolic dispersion in SnO₂."

FDTD simulation is now "FDTD simulations".

References:

- 1. McLeod, A. S. et al. Model for quantitative tip-enhanced spectroscopy and the extraction of nanoscale-resolved optical constants. *Phys. Rev. B* **90**, 85136 (2014).*
- 2. Nikitin, A. Y. et al. Real-space mapping of tailored sheet and edge plasmons in graphene nanoresonators. *Nat. Photonics* **10**, 239–243 (2016).*
- 3. Hillenbrand, R. & Keilmann, F. Complex optical constants on a subwavelength scale. *Phys. Rev. Lett.* **85**, 3029–3032 (2000).*
- 4. Neuman, T. et al. Mapping the near fields of plasmonic nanoantennas by scattering-type scanning near-field optical microscopy. *Laser Photonics Rev.* **9**, 637–649 (2015).*

=====
Reviewer #3 (Remarks to the Author):

The authors have addressed the technical objections I raised and with the extensive revisions, the manuscript looks in much better shape now. In particular, the theoretical analysis and their interpretation of experimental results now appear to be valid.

As I alluded to in my first report, I am still not convinced that the central idea of this work -- studying a new uniaxial hyperbolic material -- presents any substantial conceptual advance to merit publication in Nature Comm. Besides there is a significant overlap in the analysis as well as the experiment compared to other papers published this year (eg. Adv. Mater. 32, 1906530, 2020) although the material system studied there is hBN instead of SnO₂. Having said that, I understand that this judgement is somewhat subjective and will therefore leave it up to the editor to take a call on this point.

In summary, the manuscript now appears to be technically sound enough to be published.

We thank the reviewer for the assessment of our work including the validation of our theoretical analysis and experimental results interpretation. We appreciate that the reviewer considers our work ready for publishing.

REVIEWERS' COMMENTS

Reviewer #2 (Remarks to the Author):

The authors have addressed most of my concerns, and the manuscript is in a good shape. In particular, the technical part succeeds in explain the extent and limitations of both the experiments and numerical simulations.

I would like to point a couple of minor points:

- (i) The plural "phonons polaritons" is a quite odd one. A more correct and commonly used is "phonon polaritons"
- (ii) The discussion of cut-off frequencies should be more precise. From Fig. 3c is clear that the cut-off frequency is an upper limit (since the lowest order is the highest in frequency, and also the magnitude of the wavevector, Fig. 3d, is increasing with decreasing frequency). This is opposite to the usual cut-off frequency in dielectric waveguides, where the cut-off frequency is a lower limit of the waveguide mode dispersion. Please clarify that in the main text. It would be more visual if it is indicated in figure 4a (even in Figs. 3c).
- (iii) Additionally, indicating the phonon frequencies and names, (e.g. TO_{xx}, TO_{zz}, LO_{xx}, LO_{zz}) at its frequencies in Fig 2a, to further understand the connection between phonon frequencies and different polariton frequency ranges.

Response

We thank the reviewers for the careful assessment of our work. We have addressed all their comments and believe that these additions have improved our manuscript. Below, we provide a point-by-point reply, with the reviewers' comments in *blue italics* and our response in Black.

=====
Reviewer comments:

The authors have addressed most of my concerns, and the manuscript is in a good shape. In particular, the technical part successes in explain the extent and limitations of both the experiments and numerical simulations.

We are happy that we had covered all the concerns of the reviewer and we really appreciate the compliments. We thank the reviewer again for the availability and effort to make this work better.

I would like to point a couple of minor points:

(i) The plural "phonons polaritons" is a quite odd one. A more correct and commonly used is "phonon polaritons"

We agree with the reviewer and we had corrected all the instances to "phonon polaritons".

(ii) The discussion of cut-off frequencies should be more precise. From Fig. 3c is clear that the cut-off frequency is an upper limit (since the lowest order is the highest in frequency, and also the magnitude of the wavevector, Fig. 3d, is increasing with decreasing frequency). This is opposite to the usual cut-off frequency in dielectric waveguides, where the cut-off frequency is a lower limit of the waveguide mode dispersion. Please clarify that in the main text. It would be more visual if it is indicated in figure 4a (even in Figs. 3c).

We added the following sentence in page 10 to make this discussion point clearer:

"Notice that, in contrast to dielectric waveguides where the cutoff frequency is at the lower limit of the waveguide mode dispersion, for the SnO₂-NBs the cutoff frequency is at the upper limit since the magnitude of the transversal wavevector increases with decreasing frequency, as a consequence of the band type I dispersion of the SnO₂ HPhP modes."

Also, we now indicate these cutoff frequencies in Figures 2e and 4a of the main manuscript.

(iii) Additionally, indicating the phonon frequencies and names, (e.g. TO_{xx}, TO_{zz}, LO_{xx}, LO_{zz}) at its frequencies in Fig 2a, to further understand the connection between phonon frequencies and different polariton frequency ranges.

We appreciate the suggestion from the reviewer and we have implemented it in the manuscript. Fig. 2a is now as presented below:

The frequency values for these phonon modes were already in the Supplementary Information. However, to make this information promptly accessible to the readership, we now list these phonon frequencies and respective names in page 6 of the manuscript.